# *Lactococcus lactis'* Effect on the Intestinal Microbiota of *Streptococcus agalactiae*-Infected Zebrafish (*Danio rerio*)

Chunyan Tan,[a] Qiuyue Li,[a] Xuejiao Yang,[a] Jiayu Chen,[a] Qilin Zhang,[a] Xianyu Deng[a]

[a]Faculty of Life Science and Technology, Kunming University of Science and Technology, Yunnan, Kunming, China

Chunyan Tan, Qiuyue Li, Xuejiao Yang, and Jiayu Chen contributed equally to this article. The order of authors is determined by their contributions.

**ABSTRACT** *Streptococcus agalactiae* is a common pathogen in aquaculture that disrupts the balance of the intestinal microbiota and threatens fish health, causing enormous losses to the aquaculture industry. In this study, we isolated and screened a *Lactococcus lactis KUST48* (*LLK48*) strain with antibacterial effect against *S. agalactiae in vitro* and used it as a potential probiotic to explore its therapeutic effect on zebrafish (*Danio rerio*) infected with *S. agalactiae*. This study divided zebrafish into 3 groups: control group, injected with phosphate-buffered saline; infection group, injected with *S. agalactiae*; and treatment group, treated with *LLK48* after *S. agalactiae* injection. Then, the 16S rRNA gene sequences of the intestinal microbiota of these 3 groups were sequenced using Illumina high-throughput sequencing technology. The results showed that the relative abundance of intestinal bacteria was significantly decreased in the infection group, and a high relative abundance of *S. agalactiae* was observed. The relative abundance of the intestinal microbiota was increased in the treatment group, with a decrease in the relative abundance of *S. agalactiae* compared to that in the control group. In the Cluster of Orthologous Groups of proteins function classification, the relative abundance of each biological function in the infection group was significantly lower than that of the control and treatment groups, showing that *LLK48* has a positive therapeutic effect on zebrafish infected with *S. agalactiae*. This study provides a foundation for exploring the pathogenic mechanism of *S. agalactiae* on fish and their intestinal symbionts, and also presents a new approach for the treatment of *S. agalactiae* infections in fish aquaculture systems.

**IMPORTANCE** *L. lactis KUST48* (*LLK48*) with a bacteriostatic effect against *S. agalactiae* was isolated from tilapia intestinal tracts. *S. agalactiae* infection significantly reduced the relative abundance of intestinal bacteria and various physiological functions in zebrafish intestines. *LLK48* demonstrated infection and subsequent therapeutic effects on the *S. agalactiae* infection in the zebrafish intestine. Therefore, the potential probiotic *LLK48* can be considered as a therapeutic treatment for *S. agalactiae* infections in aquaculture, which can reduce the use of antibiotics and help maintain fish health.

**KEYWORDS** *Streptococcus agalactiae*, *Lactococcus lactis*, zebrafish, intestinal microbiota

**Ad Hoc Peer Reviewer** Cynthia Sequeiros; Mehrdad Moosazadeh Moghaddam

Address correspondence to Xianyu Deng, dengxy1008@126.com.

The authors declare no conflict of interest.

**S**treptococcus agalactiae, belonging to group B *Streptococcus*, is a facultative anaerobic Gram-positive bacterial species, which can infect both aquatic fish (1) and mammal species (2), causing inflammation (3). *S. agalactiae* has been reported to infect more than 30 fish species (4) and can cause up to 50% mortality in infected fish (5). *S. agalactiae* can survive in macrophages; it enters the blood-brain barrier and then enters the blood and central nervous system to quickly infect other organs and tissues, causing the body to develop symptoms of bacterial sepsis (6). The disease occurs mainly in spring, summer, and autumn, appearing in juvenile and adult fish. *S. agalactiae* is highly infectious, with an infection rate of 20% to 30% (7), causing a serious loss in fish breeding and production.

The abusive use of antibiotics in recent decades has resulted in a decline in the efficacy of antibiotics. Antibiotics can disrupt the balance of the normal microbiota in the intestinal tract of the body, leading to antibiotic susceptibility of bacteria (8). At the same time, antibiotics remain in the body, causing serious problems, such as allergies and ecological pollution (9, 10). As an alternative to antibiotics, probiotics can avoid the negative side effects of antibiotics and help improve nutrient absorption, alimentary canal conditions, host's rapid response to diseases, and body's immunity (11–13). Common types of probiotics are lactic acid bacteria (LAB), *Bacillus* spp., yeasts, and photosynthetic bacteria (14). For example, LAB have therapeutic effects on respiratory diseases (15), urinary system disorders (16), allergic reactions (17), and intestinal inflammation (18). Generally, native microorganisms can more easily colonize the intestinal than non-native microorganisms, and the probiotic effect lasts longer. In aquaculture, fish-derived probiotics perform better than other host-derived probiotics (19, 20). Lazado et al. (21) isolated *Pseudomonas* sp. and *Psychrobacter* sp. strains GP21 and GP12 from Atlantic cod (*Gadus morhua*). Under different physical conditions, they had an antagonistic effect on 2 key fish pathogens, *Vibrio anguillarum* NCIMB 2133 and *Aeromonas salmonicida* subsp. *salmonicida* NCIMB 1102, and had no pathogenicity or lethality to fish, indicating that these 2 strains had potential probiotic characteristics.

Probiotics can colonize the host intestine and maintain the host intestinal microbiota in a stable state. The type, distribution and quantity of intestinal microbiota affects various physiological activities of the host. Intestinal microbiota is interdependent and can respond to the stress caused by various adverse environments (22). When the host is infected, intestinal microbes maintain immune stability through diversity adjustment to control inflammation (23). The Chao and Shannon indices of intestinal bacterial species diversity of butyric acid-fed treatment piglets were higher than those of the control group, which may have contributed toward alleviating their inflammatory response (23). In addition, intestinal microorganisms produce active substances/metabolites, which can affect the digestion and intestinal immunity of the host (24).

Zebrafish have been employed to explore the colonization and immune protection capabilities of probiotics in intestines. Singer et al. used green fluorescent protein-labeled immune cells and red fluorescent-labeled pathogens to explore changes in bacterial colonization in the zebrafish intestine (25). A strictly anaerobic bacterial species, *Eubacterium limosum*, was able to successfully colonize the intestines of 5 day old juvenile zebrafish (26). In another study, zebrafish were treated with *Lactobacillus plantarum* after being exposed to pathogenic *Aeromonas hydrophila*. They found that *L. plantarum* 08.923 had strong capabilities to adhere to and colonize zebrafish intestines and upregulated the immune protection function of the epithelial barrier, which had the potential function of preventing mucosal damage caused by acute infections (27). Therefore, Zebrafish (*Danio rerio*) is a model fish used in studies of pathogen infection and host immune response. The immune response of zebrafish can be activated through changes in the intestinal microbiota (i.e., diversity and abundance) after pathogen infection (28).

In this study, we first isolated LAB with bacteriostatic activity against *S. agalactiae* and then used intraperitoneal injection to establish *S. agalactiae*-infected zebrafish models. We then tested whether a potential probiotic bacterium, *L. lactis KUST48* (*LLK48*), could be employed to reduce disease symptoms and mortality in *S. agalactiae*-infected zebrafish. This study could provide a theoretical basis for the probiotic treatment strategy of fish diseases caused by *S. agalactiae* and serve as a reference for developing alternatives to antibiotic treatments of fish diseases.

## RESULTS

**Isolation and antimicrobial activity of intestinal bacteria.** A total of 79 strains of LAB were isolated from the tilapia intestines. After the *S. agalactiae* inhibition test (these data have not been published in other journals) (Table 1), the results showed that the strain *KUST48* had a good inhibitory effect on *S. agalactiae*. The diameter of the inhibition zone of the positive control was 29 mm and that of *KUST48* was 17 mm.

**TABLE 1** Bacteriostatic results of 79 strains of LAB against *S. agalactis* (diameter of bacteriostatic circle mm)[a]

| Strain no | Inhibition zone diam (mm) | Strain no | Inhibition zone diam (mm) | Strain no | Inhibition zone diam (mm) | Strain no | Inhibition zone diam (mm) |
|---|---|---|---|---|---|---|---|
| 1 | - | 22 | - | 43 | - | 64 | - |
| 2 | - | 23 | - | 44 | - | 65 | - |
| 3 | 11 | 24 | 11 | 45 | - | 66 | - |
| 4 | 9 | 25 | 12 | 46 | - | 67 | - |
| 5 | 10 | 26 | 11 | 47 | - | 68 | - |
| 6 | 9 | 27 | 9 | 48[b] | 17 | 69 | - |
| 7 | 13 | 28 | 11 | 49 | - | 70 | - |
| 8 | 9 | 29 | 10 | 50 | 12 | 71 | - |
| 9 | 14 | 30 | 12 | 51 | - | 72 | - |
| 10 | 12 | 31 | 11 | 52 | - | 73 | - |
| 11 | 11 | 32 | 12 | 53 | - | 74 | - |
| 12 | 11 | 33 | 11 | 54 | - | 75 | - |
| 13 | - | 34 | - | 55 | - | 76 | - |
| 14 | - | 35 | - | 56 | - | 77 | - |
| 15 | - | 36 | 11 | 57 | - | 78 | - |
| 16 | 12 | 37 | - | 58 | - | 79 | - |
| 17 | - | 38 | 10 | 59 | - | negative | - |
| 18 | - | 39 | - | 60 | - | positive | 29 |
| 19 | - | 40 | 10 | 61 | - | | |
| 20 | - | 41 | - | 62 | - | | |
| 21 | - | 42 | - | 63 | - | | |

[a]"-" Represents no bacteriostatic effect.
[b]The number 48 in Table 1 is KUST48.

No inhibition zone was observed for the negative control (Fig. 1). The MIC was determined to be 125 mg/mL. The results of *KUST48* strain sequence alignment showed that the total length of the 16S rRNA sequence was 1453 bp, with 99.93% similarity to known *L. lactis* sequences in GenBank, which was confirmed by phylogenetic analysis (Fig. 2). This strain was sent to the China General Microbiological Culture Collection Center for strain preservation (CGMCC no. 20699, https://cgmcc.net/serve/jzsearch). The GenBank accession number of the 16S rDNA sequence is OK087340.

**Survival of zebrafish after *S. agalactiae* exposure.** A few zebrafish died in each of the 3 experimental groups. Zebrafish in the control group showed no signs of *S. agalactiae* infection after injection with PBS, and 2 probably died due to injury at the time of injection, resulting in a mortality rate of 6.67%. Zebrafish in the infection group all showed obvious signs of *S. agalactiae* infection; during the initial stage of infection, zebrafish show local congestion and swelling in the abdomen and bottom of the pectoral fin and then swim alone, away from the rest of the zebrafish population. During the later stages of infection, zebrafish showed exophthalmos, rotating swimming, or uncoordinated swimming. By the end of the experiment, 12 individuals died, and the death rate was 40%. The *LLK48* treatment was performed on the treatment group at 24 hpi after *S. agalactiae* injection, but the infection symptoms were milder than in the infection group, persisting until the end of the experiment. Four individuals died, resulting in a mortality rate of 13.33%.

**Analysis of sequences.** A total of 795287 effective sequences were obtained from 15 samples, with an average length of 418.31 bp. The effective sequence and sequence length range of each sample are shown in Table 2. The sequence length range of control group (219-483 bp) was more variable than that for infection group (239-432 bp) and treatment group (234-466 bp). Comparing the mean length of each group, the infection group had the longest sequence length, followed by the treatment group, and the shortest sequence length was observed in the control group. All clean sequence data of the 15 libraries were deposited in the Sequence Read Archive (SRA) database of the NCBI (Accession No.: SRX9714894-SRX9714908). The species coverage index was 0.99, indicating that the data obtained covered almost the entire bacterial microbiota of zebrafish intestines. With an increase in sequencing data, the rarefaction curve tended to approach an asymptote, indicating that the amount of sequence data

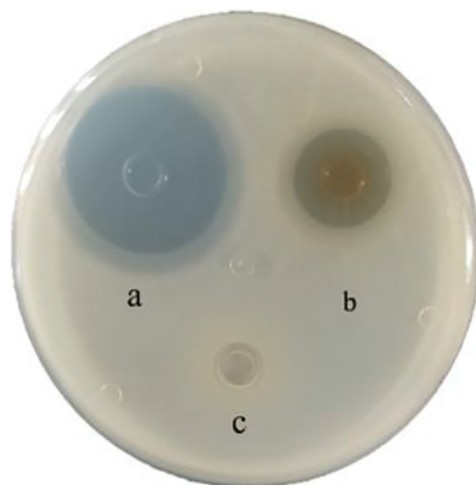

**FIG 1** Bacteriostatic effect of *KUST48* strain on *S. agalactiae*.

was sufficiently representative (Fig. 3). In the principal-component analysis, the samples of the same treatment group were clustered together, indicating a significant difference ($P < 0.05$) among the control, infected, and treatment groups (Fig. 4).

**Analysis of microbial community composition at the phylum level.** A total of 15 phyla were detected in all 15 zebrafish intestine samples. All phyla are represented in all the samples and groups (Fig. 5), including Actinobacteria, Fusobacteria, Firmicutes, and Proteobacteria. Among these, Proteobacteria was the dominant bacterial phylum in the control group, while Firmicutes was dominant in the infection group. In the treatment group, the relative abundance of Firmicutes decreased significantly ($P < 0.05$), while the relative abundance of Proteobacteria increased significantly ($P < 0.05$), which was close to the normal level of the control group.

**Analysis of microbial community composition at the genus and species levels.** When analyzing the species' relative abundance of each group, a total of 194 bacterial genera were detected among the 3 groups (Table 3). Of these, 144, 111, and 130 bacterial genera were detected in the control group, infection group, and treatment group, respectively. *Escherichia-Shigella*, and *Reyranella* were the ubiquitous genera (the sum of their relative abundance > 0.1%) in all 3 groups (Table 3). In the control group, *Ensifer* and *Shinella* were the dominant genera, with > 27% relative abundance. In the infection group, *Streptococcus* was the dominant genus, with > 94%. In the treatment group, *Streptococcus*, *Ensifer*, and *Shinella* were the dominant genera, all with relative abundances > 10%. Four genera (including *Streptococcus*, *Legionella*, *Shinella* and *Reyranella*) showed a significant difference ($P < 0.05$) among the 3 groups. Among the 6 genera with the overall highest average relative abundance, *Ensifer*, *Shinella*, *Acinetobacter*, *Bosea*, *Legionella*, and *Pseudomonas* were significantly inhibited in the infection group due to the proliferation of *S. agalactiae*. For these 6 genera in the control group and treatment group, the average relative abundance of

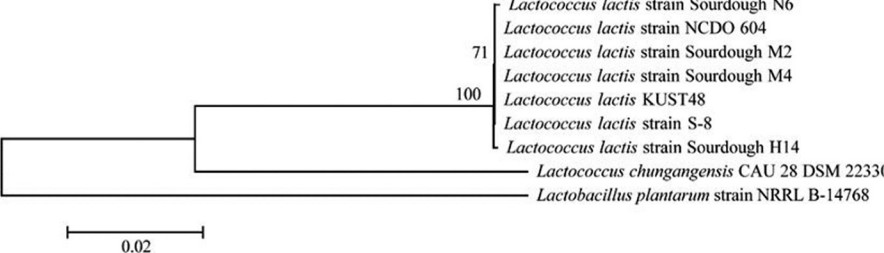

**FIG 2** The evolutionary history of *LLK*48 inferred by using the Neighbor-Joining method on MEGA 5.05 software. The bootstrap consensus tree inferred from 1000 replications.

**TABLE 2** Information statistics results of 15 samples[a]

| Sample | Seq._num. | Base_num. | Mean_length | Min._length | Max._length |
|---|---|---|---|---|---|
| a1 | 64170 | 26358289 | 410.7571918 | 245 | 431 |
| a2 | 51619 | 21152424 | 409.7798098 | 229 | 448 |
| a3 | 43326 | 17693479 | 408.3801643 | 239 | 431 |
| a4 | 48687 | 19924760 | 409.2418921 | 236 | 483 |
| a5 | 65732 | 26902937 | 409.2821913 | 219 | 431 |
| b1 | 49531 | 21210479 | 428.2263431 | 403 | 431 |
| b2 | 53365 | 22827074 | 427.7536588 | 239 | 431 |
| b3 | 47583 | 20377586 | 428.2534939 | 245 | 432 |
| b4 | 56164 | 24051066 | 428.2292216 | 402 | 430 |
| b5 | 58257 | 24936564 | 428.0440805 | 239 | 432 |
| c1 | 50956 | 21267011 | 417.3602912 | 243 | 432 |
| c2 | 46040 | 19224392 | 417.5584709 | 310 | 434 |
| c3 | 57195 | 23879299 | 417.5067576 | 242 | 466 |
| c4 | 53583 | 22372220 | 417.5245880 | 234 | 458 |
| c5 | 49079 | 20455086 | 416.7787852 | 262 | 432 |

[a]Columns 1–6 are sample-related information, followed by sample number sequence number, base number, mean length, shortest sequence length, and longest sequence length. Control groups (a1 to a5) were injected with phosphate-buffered saline; infected groups (b1 to b5) were injected with *S. agalactiae*; and treatment groups (c1-c5) were treated with LLK48 after *S.agalactiae* injection.

*Acinetobacter* was elevated while that of the other 5 genera was lower in the treatment group than in the control group. This confirms that *S. agalactiae* infection altered the relative abundance of the intestinal microbiota in the zebrafish at the genus level.

In total, 258 bacterial species were detected in all 3 groups, of which 190, 150, and 174 were detected in the control group, infection group, and treatment group, respectively. *S. agalactiae* was significantly different among the groups ($P < 0.05$), as it was caused by the respective changes in ecological niches resulting from *S. agalactiae* or *LLK48* injection. At the species level, aside from *S. agalactiae*, the bacterial species with the top 10 relative abun-

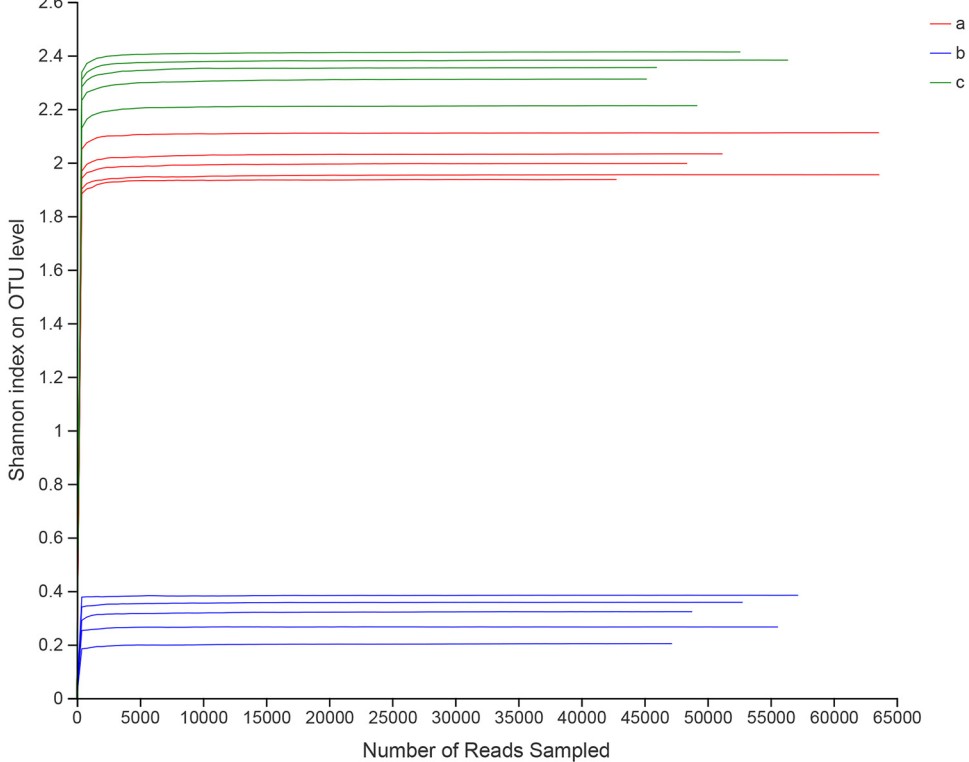

**FIG 3** Shannon index analysis of the zebrafish intestinal samples.

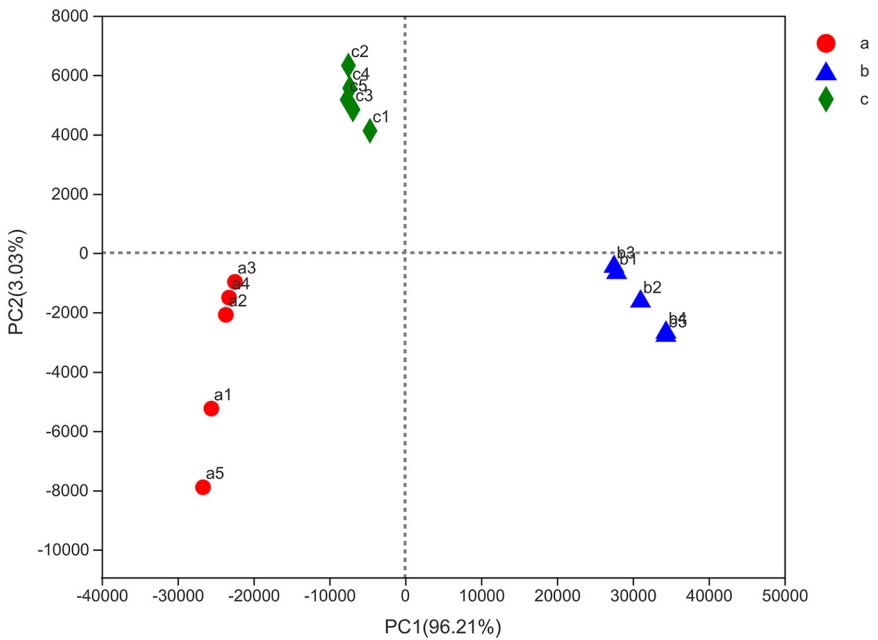

**FIG 4** Principal-component analysis (PCA) of the zebrafish intestinal samples.

dances (Table 3) were unclassified *Bosea*, uncultured *Acinetobacter*, unclassified *Legionella*, uncultured *Shinella*, unclassified *Pseudomonas*, *Ensiferadhaerens*, and *Escherichia coli*, uncultured *Legionella*, unclassified *Reyranella*, and *Agrobacterium radiobacter*.

In the comparison of relative abundance at the genus level of each group, except for the genus *Streptococcus*, the relative abundances of the control group and treatment group were similar and higher than those of the infection group. At the species level, the relative abundance of *E. coli* was the highest in the control group and the lowest in the treatment group. The relative abundance of *S. agalactiae* was also the same as *Streptococcus* at the

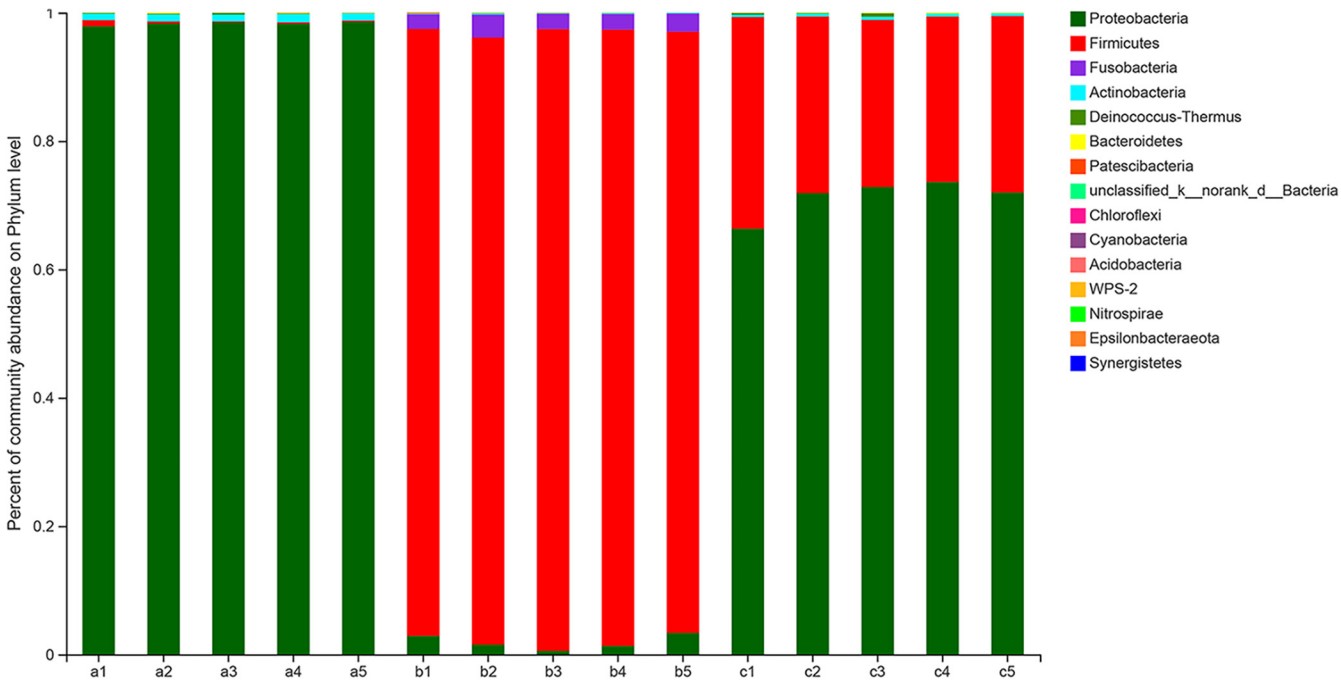

**FIG 5** Analysis of microbial community composition at the phylum level in zebrafish intestinal samples.

**TABLE 3** Relative abundances (%) of the top 10 intestinal bacterial genera and species from 3 groups

| Taxonomic categories | Control group (%) | Infection group (%) | Treatment group (%) |
|---|---|---|---|
| **Genus** | | | |
| *Streptococcus* | $0.05655 \pm 0.01832^a$ | $94.35 \pm 1.612^b$ | $27.62 \pm 2.882^c$ |
| *Legionella* | $7.335 \pm 1.731^a$ | $0.004734 \pm 0.005796^b$ | $2.428 \pm 0.2655^c$ |
| *Bosea* | $7.024 \pm 0.55681^a$ | $0.003823 \pm 0.001823^b$ | $3.844 \pm 0.1458^a$ |
| *Acinetobacter* | $6.419 \pm 1.6491^a$ | $0.02373 \pm 0.01193^b$ | $9.895 \pm 2.9671^a$ |
| *Pseudomonas* | $4.46 \pm 0.4262^a$ | $0.007077 \pm 0.006729^b$ | $2.43 \pm 0.5015^a$ |
| *Shinella* | $36.87 \pm 2.265^a$ | $0.01303 \pm 0.003364^b$ | $13.45 \pm 0.6226^c$ |
| *Ensifer* | $27.71 \pm 1.798^a$ | $0.02286 \pm 0.01581^b$ | $23.96 \pm 0.8232^a$ |
| *Escherichia-Shigella* | $1.797 \pm 1.502^a$ | $1.519 \pm 1.181^a$ | $0.5268 \pm 0.2383^b$ |
| *Reyranella* | $1.189 \pm 0.06378^a$ | $0.002043 \pm 0.002525^b$ | $0.4742 \pm 0.0443^c$ |
| Allorhizobium-Neorhobium-Pararhizobium-Rhizobium hizobium-Rhizobium | $1.088 \pm 0.0961^a$ | $0.001995 \pm 0.001518^b$ | $1.375 \pm 0.09011^a$ |
| Pannonibacter | $0.9485 \pm 0.048081^a$ | $0.003552 \pm 0.00256^b$ | $1.09 \pm 0.019021^a$ |
| **Species** | | | |
| *Streptococcus agalactiae* | $0.05253 \pm 0.016841^a$ | $94.35 \pm 1.612^b$ | $27.62 \pm 2.883^c$ |
| unclassified_*Bosea* | $7.024 \pm 0.5568^a$ | $0.003823 \pm 0.001823^b$ | $3.844 \pm 0.1458^a$ |
| Unculturedorganism_*Acinetobacter* | $6.382 \pm 1.642^a$ | $0.0204 \pm 0.0113^b$ | $9.665 \pm 2.9^a$ |
| unclassified_*Legionella* | $6.056 \pm 1.383^a$ | $0.004322 \pm 0.005156^b$ | $1.986 \pm 0.2508^c$ |
| uncultured_bacterium_*Shinella* | $36.87 \pm 2.265^a$ | $0.01303 \pm 0.003364^b$ | $13.45 \pm 0.6226^c$ |
| unclassified_*Pseudomonas* | $3.959 \pm 0.3754^a$ | $0.002181 \pm 0.001994^b$ | $0.8874 \pm 0.1466^c$ |
| *Ensiferadhaerens* | $27.71 \pm 1.798^a$ | $0.02286 \pm 0.01581^b$ | $23.96 \pm 0.8232^a$ |
| *Escherichia coli* | $1.796 \pm 1.5^a$ | $1.509 \pm 1.172^a$ | $0.5268 \pm 0.2383^b$ |
| uncultured_bacterium_*Legionella* | $1.279 \pm 0.3811^a$ | $0.0004121 \pm 0.0009216^b$ | $0.442 \pm 0.01854^c$ |
| unclassified_*Reyranella* | $1.189 \pm 0.06378^a$ | $0.002043 \pm 0.002525^b$ | $0.4742 \pm 0.0443^c$ |
| *Agrobacterium radiobacter* | $1.041 \pm 0.09499^a$ | $0.001646 \pm 0.001769^b$ | $1.339 \pm 0.08473^a$ |

[a,b,c]Different superscripts indicate significant differences between groups $P < 0.05$. Control group was injected with phosphate-buffered saline; infected group was injected with *S. agalactiae*; and treatment group was treated with LLK48 after *S. agalactiae* injection.

genus level. The relative abundance of *S. agalactiae* in the control group was extremely low ($0.05253 \pm 0.016841$). The relative abundance of the infection group was $94.35 \pm 1.612$, and the relative abundance of the treatment group was significantly reduced to $27.62 \pm 2.883$.

**Diversity analysis.** A total of 114, 66, and 120 species were detected in the control group, infection group, and treatment group, respectively. The number of species of infection group was less than that of the other 2 groups, and the number of species of control group and treatment group was similar. The results of each alpha diversity index showed that the species diversity of the infection group was significantly reduced ($P < 0.05$) compared to those of the control group and treatment group, which were comparatively similar (Table 4). The Sob, Shannon, ACE, and Chao indices showed the following trend for the 3 groups: treatment group > control group > infection group (Table 4).

**Cluster of Orthologous Groups of protein function classification results.** PICRUSt was used to predict the functional composition of the microbial community, and a total of 23 annotated functions were obtained (Table 5). The top 10 functional relative abundance values were successively amino acid transport and metabolism; general function prediction only transcription; carbohydrate transport and metabolism; inorganic ion transport and metabolism; energy production and conversion; cell wall/

**TABLE 4** $\alpha$ diversity analysis of each group

| Items | Control group | Infection group | Treatment group |
|---|---|---|---|
| Observed species | $114 \pm 9$ | $66 \pm 11$ | $120 \pm 5$ |
| Sob | $123.2 \pm 13.16^a$ | $69 \pm 13.17^b$ | $128.6 \pm 5.86^a$ |
| Shannon | $2.00 \pm 0.07^a$ | $0.31 \pm 0.07^b$ | $2.33 \pm 0.08^c$ |
| Ace | $153.58 \pm 22.13^a$ | $118.48 \pm 26.66^b$ | $154.60 \pm 7.75^a$ |
| Chao | $149.81 \pm 25.35^a$ | $96.03 \pm 21.30^b$ | $153.46 \pm 9.80^a$ |
| Coverage | 0.99 | 0.99 | 0.99 |

[a,b,c]Different superscripts indicate significant differences between groups $P < 0.05$. Control group was injected with phosphate-buffered saline; infected group was injected with *S. agalactiae*; and treatment group was treated with LLK48 after *S. agalactiae* injection.

**TABLE 5** Statistical results of COG function classification

| Category | Control group | Infection group | Treatment group | Description |
|---|---|---|---|---|
| 1 | 36418800[a] | 8514291[b] | 25771954[c] | Function unknown |
| 2 | 35403161[a] | 7800647[b] | 24681514[c] | Amino acid transport and metabolism |
| 3 | 28367753[a] | 7905292[b] | 20344440[c] | General function prediction only |
| 4 | 26815821[a] | 6624626[b] | 19284172[c] | Transcription |
| 5 | 25638417[a] | 8636276[b] | 17819817[c] | Carbohydrate transport and metabolism |
| 6 | 23088580[a] | 4332632[b] | 16002245[c] | Inorganic ion transport and metabolism |
| 7 | 22356939[a] | 3229417[b] | 15397363[c] | Energy production and conversion |
| 8 | 18427779[a] | 5063880[b] | 13015581[c] | Cell wall/membrane/envelope biogenesis |
| 9 | 16514229[a] | 2616593[b] | 11839706[c] | Signal transduction mechanisms |
| 10 | 16108838[a] | 5881761[b] | 11526951[c] | Replication, recombination, and repair |
| 11 | 13929274[a] | 7685245[b] | 10871179[c] | Translation, ribosomal structure, and biogenesis |
| 12 | 13830425[a] | 2102318[b] | 9778181[c] | Lipid transport and metabolism |
| 13 | 12009303[a] | 2914783[b] | 8428149[c] | Posttranslational modification, protein turnover, chaperones |
| 14 | 10782653[a] | 2228693[b] | 7388106[c] | Coenzyme transport and metabolism |
| 15 | 8844528[a] | 597647[b] | 5884183[c] | Secondary metabolites biosynthesis, transport, and catabolism |
| 16 | 7053330[a] | 3624617[b] | 5414032[c] | Nucleotide transport and metabolism |
| 17 | 6762522[a] | 1453009[b] | 4761423[c] | Intracellular trafficking, secretion, and vesicular transport |
| 18 | 4740124[a] | 296647[b] | 3247958[c] | Cell motility |
| 19 | 4003369[a] | 2555117[b] | 3254121[c] | Defense mechanisms |
| 20 | 2951577[a] | 1230877[b] | 2124442[c] | Cell cycle control, cell division, chromosome partitioning |
| 21 | 105633[a] | 1658[b] | 76556[c] | Chromatin structure and dynamics |
| 22 | 48413[a] | 1041[b] | 43255[a] | RNA processing and modification |
| 23 | 39230[a] | 126[b] | 29328[c] | Cytoskeleton |
| 24 | 34853[a] | 594[b] | 12046[c] | Extracellular structures |

[a,b,c]Different superscripts indicate significant differences between groups $P < 0.05$. Control group was injected with phosphate-buffered saline; infected group was injected with *S. agalactiae*; and treatment group was treated with LLK48 after *S. agalactiae* injection.

membrane/envelope biogenesis; signal transduction mechanisms; replication, recombination and repair; translation, ribosomal structure, and biogenesis. Under each function, there were significant differences among the control group, infection group, and treatment group ($P < 0.05$).

## DISCUSSION

The diversity and relative abundance of intestinal microbes are important indicators for host health (22, 23, 29, 30). Probiotics, pathogens, and viruses in the intestine constitute a dynamic micro-ecosystem. When the intestinal microbiota is maladjusted or the host is infected with a disease, it leads to a reduction in intestinal microbial diversity, disruption of ecological balance, and an increase in the number of pathogens. Studies have shown that probiotics can inhibit or buffer the reduction of intestinal bacterial diversity and inhibit the growth of pathogens (9).

In this study, *S. agalactiae* inhibited the growth of other bacteria in the zebrafish intestine, and as a probiotic, *LLK48* could inhibit or buffer the harm caused by *S. agalactiae* to the intestinal bacteria. In a previous study, it was found that the Shannon, Simpson, ACE, and Chao indices of intestinal bacteria were significantly reduced in zebrafish infected with *S. agalactiae* (29). This evidence indicated that *S. agalactiae* infection decreased the diversity of intestinal microbes, while *LLK48* reduced this effect to a certain degree. In the infection group, *S. agalactiae* infection resulted in higher mortality and significant signs of disease in zebrafish, such as exophthalmos, rotating swimming, or uncoordinated swimming, which were similar to the disease symptoms observed in our pre-experimental study (31, 32). However, in the treatment group, treatment with *LLK48* significantly reduced mortality and symptoms of disease in the zebrafish. In addition, the results of the alpha diversity index in this study are consistent with those of a previous study (29, 33), demonstrating the reliability of the results of this study and confirming that *S. agalactiae* infection decreases the bacterial diversity of zebrafish intestines. *LLK48* treatment restored the bacterial diversity of zebrafish intestines, which suggests that this approach could be used to treat *S. agalactiae*-borne infection in zebrafish.

The results in this study found a minimal quantity (relative abundance of 0.05%) of *S. agalactiae* in the intestine of healthy zebrafish. Once the healthy zebrafish were infected with *S. agalactiae*, this pathogenic intestinal bacterium increased rapidly, inhibiting the growth of other bacteria. *LLK48* significantly inhibited *S. agalactiae* proliferation in the zebrafish intestine and buffered the adverse effects of *S. agalactiae* on other bacteria. *L. lactis* is one of the most commonly used probiotics in aquaculture (34, 35). Dong et al. used *L. lactis* 16-7 to treat crucian carp infected with *Aeromonas hydrophila*, finding that this strain could initiate immune regulation and inhibit an inflammatory response (36). The *L. plantarum* used by Lin et al. also had a therapeutic effect on hippocampal enteritis (37).

In this study, we found, for the first time, that *LLK48* had a strong bacteriostatic effect against *S. agalactiae in vitro*. We found that the relative abundance of *S. agalactiae* was significantly reduced in the treatment group compared to the infection group, indicating that *LLK48* could effectively inhibit *S. agalactiae in vitro* and *in vivo,* and reduce the relative abundance of *S. agalactiae* in the intestinal tract of zebrafish *in vivo*. These results indicated that *LLK48* could be an effective probiotic for the treatment of *S. agalactiae* infections in fish.

Probiotics enhance the intestinal mucosal barrier function (30), prevent the adhesion and colonization of pathogenic bacteria (38), improve the sensitivity of the immune system (39), secrete substances, and change the intestinal environment (40). The results of the Cluster of Orthologous Groups of protein (COG) function classification of this study showed that the injection of *S. agalactiae* into healthy zebrafish significantly reduced all functional relative abundance values. The injection of *LLK48* improved functional relative abundance values close to the normal level, indicating that after *S. agalactiae* entered the zebrafish's body, it affects various activities of the body, reduces various physiological functions, and may ultimately cause death. After entering the body, *LLK48* inhibits the growth of *S. agalactiae*, buffering the decline of various physiological functions. Xia et al. fed larval Nile tilapia with *Lactobaciullus rhamnosus* and *L. lactis* and found a large quantity of *L. lactis* in the intestines during feeding. However, *L. lactis* could not be detected after feeding had stopped for 1 week (41). Therefore, *L. lactis* plays a therapeutic role by competing for adhesion sites with *S. agalactiae* in the intestine. However, the molecular and/or physiological mechanisms of *L. lactis* that inhibit and reduce the relative abundance of *S. agalactiae* need to be investigated in the future.

**Conclusion.** In summary, we isolated a strain of *LLK48* from tilapia with *in vitro* antibacterial effect on *S. agalactiae*. Zebrafish were used as an experimental subject to study the *in vivo* antibacterial effect of *LLK48*. Using Illumina sequencing technology, we explored the therapeutic effect of *LLK48* as a potential probiotic on zebrafish infected with *S. agalactiae* from the perspective of intestinal bacteria. The results showed that *S. agalactiae* infection significantly reduced the relative abundance and various physiological functions of zebrafish intestinal bacteria. Furthermore, *LLK48* exhibited inhibitory and therapeutic effects on *S. agalactiae* infections in zebrafish intestines. However, the concentration of *LLK48* that is the most effective as a therapeutic treatment and the therapeutic mechanism of *LLK48* against *S. agalactiae* infections in zebrafish need to be investigated further.

## MATERIALS AND METHODS

**LAB isolation from tilapia intestinal tract.** To source *L. lactis KUST48* (*LLK48*) strains, 3 healthy live adult tilapia (*Oreochromis niloticus*) were purchased from a supermarket in Chenggong, Kunming City, China. They were placed on ice for cold anesthesia. Then, the surface of the fish body was disinfected using 75% alcohol (TianGen), and the intestinal tract was removed in a sterile console. The intestine was repeatedly rinsed using 0.9% normal saline solution, and the intestinal contents and intestinal rinses were homogenized and used as stocks. We added 50 mL stock solution to 200 mL of *Lactobacillus* enrichment medium Man Rogosa Sharpe (MRS) (HuanKai Microbial), and incubated the mixture for 24 h at 37°C on a constant temperature shaking table (Yiheng) at 150 rpm. The mixture was diluted at a concentration gradient of $10^{-1}$ to $10^{-8}$, after which we used 100 $\mu$L of each dilution of $10^{-3}$ to $10^{-7}$ and spread it on the LAB selection medium (MRS medium containing 2% calcium carbonate) at 37°C for 24 h. Subsequently, single colonies were picked and streaked repeatedly until pure cultures were obtained.

**Screening of strains with antimicrobial activity.** The Oxford cup double-plate method was used to screen the strains with antibacterial activity (42). *S. agalactiae* used in this study was preserved in the

Phage and Intestinal Microbiology research group, Prior to their infection with *S. agalactiae* strain SAM 12 belonging to serotype III, sequence type (ST)-17 from the College of Marine Sciences, Qinzhou University, Qinzhou, China. *S. agalactiae* used in this study was grown and cultured as described by Patterson et al. (43). *S. agalactiae* was activated in solid Luria-Bertani broth (LB broth) medium and cultured overnight at 37°C. Then, a single colony was selected and inoculated in liquid LB broth for subsequent experiments. The purified LAB and *S. agalactiae* were cultured to the logarithmic growth phase ($10^8$-$10^9$ CFU/mL). The LAB were centrifuged at 6000 rpm at 4°C for 10 min, and the supernatant of the bacterial solution was filtered through a filter membrane (0.22 $\mu$m) (44). *S. agalactiae* broth (2 mL) was added to 100 mL LB broth (without sugar) semi-solid medium (HuanKai Microbial, 0.75% agar concentration) at 50°C, shook gently, and poured onto the plate. Then, a sterile Oxford cup (diameter of 8 mm) was placed on it, and 200 $\mu$L of LAB supernatant was added to the Oxford cup. Meanwhile, sterile distilled water and 100 mg/mL Kanamycin (Bio FROXX) were negative and positive controls, respectively. Subsequently, the plate was placed at 4°C for 4 h to spread and incubated at 37°C for 15 h to measure the diameter of the inhibition zone (45). The diameter of the inhibition zone indicated the bacteriostatic activity of the LAB supernatants on *S. agalactiae*. Each experiment was independently repeated three times. A strain of *KUST48* with good antibacterial effects against *S. agalactiae* was selected for subsequent experiments.

The MIC of *KUST48* supernatant over *S. agalactis* was performed by the standard broth dilution method, following the previously reported methods, with minor modifications (46, 47). Briefly, the supernatant of *KUST48* was 2-fold diluted in LB broth, 100 $\mu$L of the *KUST48* supernatant (at different dilutions), along with 100 $\mu$L of *S. agalactis* ($10^6$ CFU/mL) were added to a 96-well plate. After thoroughly mixing, they were cultured at 37°C for 24 h and the absorbance at 600 nm using a microplate reader was measured. Blank controls comprised only LB broth medium and supernatant at different concentrations. Negative controls comprised only the liquid of *S. agalactis* cultures. The MIC was defined as the lowest concentration of *KUST48* supernatant in which growth of *S. agalactis* was inhibited.

**Molecular identification of bacterial strains.** In order to identify the *KUST48* with good antibacterial activity against *S. agalactiae*, total DNA was extracted using a DNA Extraction Kit (Solarbio), and 27-F (5′-AGAGTTTGATCCTGGCTCAG-3′) and 1492-R (5′-TACGGYTACCTTGTTACGACTT-3′) (48) universal primers were used for PCR amplification of 16S rRNA gene sequencing. The PCR mix (20 $\mu$L) contained 10 $\mu$L 2 × *Taq* PCR Starmix (GenStar), with 1 $\mu$L forward primer, 1 $\mu$L reverse primer, and 2 $\mu$L extracted DNA. PCR was performed using a PCR amplification system (BIOER) under the following conditions: pre-denaturation at 95°C for 10 min, followed by 30 cycles of denaturation at 95°C for 30 s, annealing for 1 min, and extension at 72°C for 1 min. The final cycle was followed by a 7 min extension at 72°C. The quality of PCR amplicons was detected by 1% agarose gel electrophoresis. The PCR products were sent to Sangon Biotech Co., Ltd. (Shanghai, China) for nucleic acid sequence detection. The detected 16S rRNA gene sequence was searched in GenBank of NCBI (https://www.ncbi.nlm.nih.gov/). The *L. lactis* (*LLK48*) standard strain and other high-scoring bacterial 16S rRNA sequences were downloaded from GenBank for phylogenetic analysis. MEGA 5.05 was used to determine the species of the strain.

**Zebrafish and experimental design.** Zebrafish were treated in accordance with the recommendations from the Guide for the Care and Use of Laboratory Animals. The experimental protocol was approved by the Ethics Committee of Research of Kunming University of Science and Technology. Wild-type (AB strain) adult individual zebrafish were purchased from the China Zebrafish Resource Center (CZRC) (http://en.zfish.cn/) (mean weight: 0.30 ± 0.05 g, average body length: 3.4 ± 0.5 cm). Healthy zebrafish were acclimatized for approximately 5 days to empty their digestive system contents using filtered freshwater.

Patterson et al. used $10^6$ CFU/mL *S. agalactiae* as the highest concentration in their experiments and found that the intraperitoneal route of infection caused the induction of a host inflammatory immune response in the adult zebrafish brain (43). A 50% survival rate was reported for zebrafish at 24 h post-injection (hpi), indicating that a $10^6$ CFU/mL concentration of *S. agalactiae* could successfully induce the immune response in the whole body of adult zebrafish before 24 hpi (43). According to previous experimental studies in our laboratory, the half lethal concentration (LC50) of *S. agalactiae* for zebrafish is $10^6$ CFU/mL (29). Thus, this concentration was employed to trigger the immune response of zebrafish in downstream experiments.

Thirty zebrafish in each group (control group, infection group, and treatment group) were then randomly divided into 3 special aquarium divider boxes (dimensions: L × W × H = 30 cm × 15 cm × 20 cm; purchased from Aquatic Animal Supplies Store in Chenggong District, Kunming) for rearing, with the aquarium temperature set at 28 ± 0.5°C and the pH maintained at 6.9–8. For the infection and the treatment groups, 10 $\mu$L ($10^6$ CFU/mL) *S. agalactiae* in phosphate-buffered saline (PBS) was injected into each of the 30 zebrafish enterocoelia, following our previous method (29). For the control group, the same volume of filtered PBS buffer was injected without *S. agalactiae*. At 24 hpi, control and infection groups were injected with 10 $\mu$L PBS, while treatment group was injected with 10 $\mu$L ($10^6$ CFU/mL) *LLK48* in PBS. Before injection, tricaine (0.02%) (Sigma-Aldrich) was used to reduce fish activity. After injection, the mortality of zebrafish was recorded daily for 120 h (5 days). These experimental processes were independently conducted three times as 3 biological replicates (in total, 90 fish individuals were used for this study).

**Zebrafish intestinal sampling.** At 120 h, 15 zebrafishes were randomly selected from each group of experimental zebrafish, i.e., placed on ice for cold anesthesia. Then, intestines of 15 zebrafish individuals in each group were collected with scalpels and tweezers, and 5 intestines were randomly selected from the intestines collected from each group and used as sequencing samples and labeled as a1 to a5 for the control group, b1 to b5 for infection group, and c1 to c5 for treatment group. All samples were prepared and stored at −80°C for later use.

**DNA extraction and PCR amplification.** Intestinal DNA was extracted using a TIANamp Stool DNA Kit (TIANGEN Biotech), according to the manufacturer's instructions. The final DNA concentration and purification were determined using a Nano Drop 2000 UV-vis spectrophotometer (Thermo Scientific), and DNA quality was checked by 1.5% agarose gel electrophoresis. The V3-V4 hyper-variable regions of the bacterial 16S rRNA gene were amplified with primers 338-F (5′-ACTCCTACGGGAGGCAGCAG-3′) and 806-R (5′-GGACTACHVGGGTWT CTAAT-3′) by the thermocycler PCR system (Applied Biosystems). The PCRs were implemented using the following steps: 3 min of denaturation at 95℃, 27 cycles of 30 s at 95℃, 30 s for annealing at 55℃, 45 s for elongation at 72℃, and a final extension at 72℃ for 10 min. PCR was performed in triplicate in a 20-$\mu$L mixture containing 4 $\mu$L of 5 × FastPfu Buffer, 2 $\mu$L of 2.5 mM dNTPs, 0.8 $\mu$L of each primer (5 $\mu$M), 0.4 $\mu$L of FastPfu polymerase, and 10 ng of template DNA. The PCR products were purified using the AxyPrep DNA Gel Extraction Kit (Axygen Biosciences) and then quantified using QuantiFluor-ST (Promega, Madison), according to the manufacturer's protocol.

**Illumina sequencing and processing of sequencing data.** To assess the effect of *LLK48* treatment on the intestinal microbiota of zebrafish, 16S rRNA gene sequences were sequenced for zebrafish intestinal bacteria in the control group, infection group, and treatment group using Illumina high-throughput sequencing technology. Purified amplicons were pooled in an equimolar and paired-end sequence (2 × 300 bp) on an Illumina MiSeq platform (Illumina, San Diego), according to the standard protocols of Majorbio Bio-Pharm Technology Co., Ltd. Raw fastq files were quality-filtered by Trimmomatic and merged by FLASH with the following criteria: (i) The reads were truncated at any site receiving an average quality score < 20 over a 50 bp sliding window; (ii) sequences longer than 10 bp were merged according to their overlap, with a mismatch of no more than 2 bp; and (iii) sequences of each sample were separated according to barcodes (exactly matching) and primers (allowing 2 nucleotide mismatches), and reads containing ambiguous bases were removed.

Operational taxonomic units (OTUs) were clustered with 97% similarity cut-off using UPARSE (version 7.1, http://drive5.com/uparse/), with a novel greedy algorithm that performed chimaera filtering and OTU clustering simultaneously. The taxonomy of each 16S rRNA gene sequence was analyzed by the RDP Classifier algorithm (http://rdp.cme.msu.edu/) against the Silva (SSU123) 16S rRNA bacterial database using a confidence threshold of 70%.

**Statistical analyses and comparison of microbial communities.** 16S rRNA amplicon sequencing was performed on the intestinal microbiota of the control, infection, and treatment groups to detect changes in the composition and relative abundance of the intestinal microbial community. A one-way analysis of variance (ANOVA) test was used to detect significant differences in index values among the 3 groups using ACE, Chao, Shannon, and Sob indices. According to the community composition data obtained, a Kruskal-Wallis test was used to detect differences in species richness between the microbial communities of the 3 groups. A *t* test was used to compare the 2 groups in the stats package of R and the SciPy package of Python. A *P* value of < 0.05 among groups was considered statistically significant.

**Functional changes in the intestinal microbial community (PICRUSt).** Finally, to reveal the changes in the primary functions of the intestinal bacterial microbiota before and after *S. agalactiae* infection and *LLK48* treatment, functional changes in the intestinal bacteria among the 3 experimental groups were compared. The PICRUSt function prediction method was used to predict the function of each group of microbial communities. The OTU abundance table was normalized by PICRUSt (PICRUSt software stores the COG information and KEGG Orthology (KO) information corresponding to Greengene IDs), i.e., the 16S marker gene in the species genome was removed. Then, we obtained the COG family information and KO information for each OTU by the corresponding Greengene ID of each OTU, and calculated the abundance of each COG and KO abundance. Based on the information of the COG database, the descriptive information of each COG and its functional information can be parsed from the eggNOG database to obtain the functional abundance spectrum; based on the information of KEGG database, the KO, pathway, and EC information can be obtained, and the abundance of each functional category can be calculated based on the OTU abundance. In addition, PICRUSt can be used to obtain information on 3 levels of metabolic pathways for pathway, and the abundance table of each level can be obtained. The experimental data were analyzed by one-way ANOVA using IBM SPSS Statistics 25. The significant difference between groups was determined by Duncan test, and *P* < 0.05 was used as the criterion for significant difference.

## ACKNOWLEDGMENTS

This work was supported by the Natural Science Foundation of China (31640079).

All authors declare no potential conflicts of interest.

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
