## [Reviewer comments · Microbiology Spectrum]

Microbiology Spectrum

Lactococcus lactis effect on the intestinal microbiota of Streptococcus agalactiae infected zebrafish (*Danio rerio*)

Chunyan Tan, Qiuyue Li, Xuejiao Yang, Jiayu Chen, Qilin Zhang, and Xianyu Deng

Corresponding Author(s): Chunyan Tan, Kunming University of Science and Technology

Review Timeline:

Submission Date:	March 28, 2022
Editorial Decision:	May 18, 2022
Revision Received:	July 16, 2022
Editorial Decision:	August 12, 2022
Revision Received:	August 25, 2022
Accepted:	September 1, 2022

Editor: Konstantinos Kormas

Reviewer(s): Disclosure of reviewer identity is with reference to reviewer comments included in decision letter(s). The following individuals involved in review of your submission have agreed to reveal their identity: Cynthia Sequeiros (Reviewer #1); Mehrdad Moosazadeh Moghaddam (Reviewer #3)

Transaction Report:

DOI: <https://doi.org/10.1128/spectrum.01128-22>

May 18, 2022

Dr. Chunyan Tan
Kunming University of Science and Technology
Yunnan , Kunming
China

Re: Spectrum01128-22 (Lactococcus lactis treatment of Streptococcus agalactiae infections affecting the structure of intestinal microbiota in zebrafish (Danio rerio))

Dear Dr. Chunyan Tan:

Link Not Available

Sincerely,

Konstantinos Kormas

Journals Department
Reviewer comments:

Reviewer #1 (Comments for the Author):

Overall Recommendation:

The authors isolated a strain of *Lactococcus lactis* KUST48 from tilapia with in vitro antimicrobial activity against *Streptococcus agalactiae*. They studied the effect of *L. lactis* strain on zebrafish infected with *S. agalactiae* from the perspective of the intestinal microbiota. The in vivo experimental test was divided into three groups: the control group (A) injected with PBS, the infected group (B) injected with *S. agalactiae*, and the treated group (C) injected with *S. agalactiae* and 24h post-injection with *L. lactis* KUST48. Then, the 16S rRNA gene sequences of the intestinal microbiota of these three groups were sequenced using Illumina high-throughput sequencing technology.

The topic of this study is very interesting. The results of this study showed that the injection of *L. lactis* KUST48 produced benefits on zebrafish infected with *S. agalactiae*. However, the authors refer to *L. lactis* as a probiotic when this was not yet demonstrated (see Abstract review). Moreover, the term "treatment" referred in the Title would not be appropriate since a more comprehensive evaluation is needed, where different doses, times, and effects of *L. lactis* on infected zebrafish are tested or evaluated. On the other hand, the manuscript was hard to read, the authors should find a native speaker to polish the English. For a better understanding of the effect of *L. lactis* KUST48 on the zebrafish intestinal microbiota, another control group with the *L. lactis* KUST48 and without the *S. agalactiae* would be useful to add.

Title:

- I suggest the following title: *Lactococcus lactis* effect on the intestinal microbiota of *Streptococcus agalactiae* infected zebrafish.

Abstract

- The abstract is very poorly written and needs to be rewritten.
- Lines 39-41. In importance section "...The probiotic *Lactococcus lactis* (LLK48) can be considered as a therapeutic and water quality modifying agent for *Streptococcus agalactiae* (SA) infections ...".
- It is not appropriate to speak of "probiotic" (throughout the manuscript), because both, other tests are needed to confirm that a bacterial strain can be considered a probiotic (i.e. antibiotic resistance genes, haemolytic activity, bile tolerance, etc.), and the intraperitoneal injection is not the proper administration way. A probiotic is administered with the food or in the rearing water. Thus, they should change "probiotic" by "potential probiotic".
- In the present study, the effect of *L. lactis* (LLK48) on water quality was not demonstrated nor evaluated. Please check this.

Keywords: Use different keywords that do not present in the title to facilitate searching on your paper

Introduction

- The Introduction is very poorly written and needs to be rewritten. Some examples: lines 46-49; 77-78; 81-87; 88-91; etc.
- Line 60, the bibliographic citation (6) is not adequate. Please use a proper citation.
- Lines 64-66. Bibliographic citation (9) does not support the written sentence.
- Lines 60-63. The bibliographic citation is not adequate. There are several papers about of the beneficial uses of probiotics in Aquaculture. You must use an appropriate citation.
- Lines 74-76: this sentence is confusing; please indicate if these data are being published in another journal or are part of this work.
- Lines 108-118. This information should be in materials and methods section.

Materials and methods:

Section 2.1

- Please change the title: "Isolation of intestinal flora" by "LAB isolation from tilapia intestinal tract".
- Please add the proportion of the dilutions performed for the isolation assay.

Section 2.2: in this section there are several mistakes. Please check it carefully

- I suggest changing the title: "Screening of antibacterial strains" by "Screening of strains with antimicrobial activity".
- Please add the origin and culture conditions of the *S. agalactiae* strain.
- Please add the dilutions performed for the minimum inhibitory concentration (MIC) assay.
- Line 141: please clarify if it is broth or semi-solid medium, if it is semi-solid medium, indicate the agar concentration.
- Lines 142: does the plate really invert to mix?
- Line 145: Please add Kanamycin concentration.

Section 2.3:

- Line 152-153: Please review the content of the following sentence: "In order to select a lactic acid bacterial strain with good antimicrobial activity against *S. agalactiae*, total DNA was extracted using a DNA Extraction Kit..."
- Line 155: "...of 16S gene sequencing...", please replace with "...of 16S rRNA gene sequencing..."
- Line: 165-167: Please review the following sentence: "The sequenced 16S was searched in GenBank of the National Center for Biotechnology Information (NCBI) (<https://www.ncbi.nlm.nih.gov/>)."

Section 2.4:

- I suggest changing the title for "Zebrafish and experimental design"
- Lines 178-181: The content of these lines should be in section 2.2.
- Line 192: "Thirty zebrafish in each group were then randomly divided into three special aquarium divider box...". Please explain which groups you are referring to.

Section 2.5

- In lines 210-212 the text is confusing: "...the intestines of 15 zebrafish were randomly divided into 5 portions, and each portion of intestine was mixed into one sequencing sample for each group". Please rewrite this sentence and explain better how each sample was prepared.

- In materials and methods, please indicate the period of time used to calculate the mortality rate

In materials and methods, please add a new section that details the prediction method of the functional changes of the intestinal microbial community (PICRUSt).

Section 2.8

- Please add the statistical analysis of the predicted functions using PICRUSt.

RESULTS

Section 3.1:

- Please change the title: "Isolation and bacteriostatic experiment of intestinal bacteria" by "Isolation and antimicrobial activity of intestinal bacteria".
- Throughout the manuscript, in the objective of this study (lines 103-104), in section 2.1 of materials and methods and in the title of section 3.1 of results, several isolated bacteria are mentioned (several isolates belonging to the group of LAB), however the result of only one is shown. I suggest adding a table with the results of antimicrobial activity against *S. agalactiae* of the 79 LAB strains isolated from tilapia intestine. Please add also the results of the minimum inhibitory concentration (MIC) described in section 2.2 of materials and methods.
- In figure legends, for Figure 1, it would be appropriate to change the name of the strain "LLK48" to "strain KUST48" since at this point the strain has not yet been identified.
- Line 269: there are several mistakes in grammar and sentence construction. Please check it carefully. Moreover, please change "...the 16S rDNA sequence..." by "...16S rRNA sequence..."

Section 3.2:

- In this section (lines 276-287) there are several mistakes in grammar and sentence construction. Please check it carefully.

Section 3.3

- Please change the title: "Optimization analysis of high-throughput sequencing data" by "Analysis of sequences".

Section 3.4

- Lines 306-307. Please, check "Actinobacteria and Fusobacteria" are repeated.
- There are several grammatical mistakes in this paragraph. Please check it carefully

Section 3.5

- Lines 316. "Escherichia-Shigella and Aeromonas were the main genera (the sum of the relative abundance > 0.1%) in all the three groups (Table 2)". Please check, in table 2 the Aeromonas genus does not appear.
- There are several mistakes in grammar and sentence construction. Please check it carefully.

Section 3.6

- Please check Table 3, the information mentioned in lines 355-359 does not appear.
- Lines 360-366 are repeated.

Section 3.7

- Lines 369-370. "The analysis results of species relative abundance in each group at the genus level are shown in Figure 5". Please check this sentence, in Figure 5 the analysis of microbial community composition at the phylum level is showed.
- The results in the section 3.7 are similar to that shown in the section 3.5, please unify the results in a single section.
- The results are very poorly presented. The manuscript has many grammatical mistakes, making it very difficult to read.
- It would be interesting to analyze the relative abundance of specific bacterial genera such as *Vibrio* sp., *Lactococcus* sp., for the different groups

Section 3.8

- Please indicate the significant differences in predicted microbial functions among the three groups. This section was poorly written. Please rewrite it.

Table 2:

- "Different superscripts identify significant differences ($P < 0.05$) in relative abundance". Please indicate whether the significant differences are within the same group or between groups.

Table 3:

- Please indicate the meaning of the categories (A, B, C, etc.) and the values that appear in Table 3. Indicate the results of the statistical analysis among 3 groups.

Figure legends

- Fig. 1: Please, change "Bacteriostatic effect of LLK48 on *S. agalactiae*. (a) Positive control: Antibacterial effect of Kanamycin on

S. agalactiae; (b) Bacteriostatic effect of LLK48 on S. agalactiae; (c) Bacteriostatic effect of sterile distilled water on S. agalactiae."; by "Bacteriostatic effect of KUST48 strain on S. agalactiae. (a) Positive control: Kanamycin; (b) KUST48 supernatant; (c) Negative control: sterile distilled water."

- Fig 3: Please, change the figure title and expand the description. The title may be: Shannon index analysis of the zebrafish intestinal samples. Add details to the legend of figure 3. Moreover, it remains to explain the meaning of a, b and c.
- Fig 4: Please, change the figure title, it may be: "Principal component analysis (PCA) of the zebrafish intestinal samples". Add the meaning of a, b and c.
- Fig 5: Please, change the figure title, it may be: "Analysis of microbial community composition at the phylum level in zebrafish intestinal samples". Describe the meaning of a1, a2, a3, ... etc
- Fig 6: This figure provides the same information as figure 5, I suggest removing it.

Discussion

-Lines 421-422: Please cite articles related to aquaculture probiotics

Reviewer #2 (Comments for the Author):

The scientific content of this manuscript is sound and experimental observations are useful. Minor errors are mostly formatting issues, although a moderate number of other corrections following careful editorial proofreading are strongly recommended. This is not a complete editorial guide but here are some examples:

- lines 2,3 species name is not properly formatted
- lines 8,9 addresses should be on separate lines
- line 33 "tilapia" should not be capitalized (it is not a proper noun)
- line 42 ineffective keywords
- line 83 replace indexes with indices
- lines 87 88, indentation is inconsistent
- line 99 correct species nomenclature
- line 97 methods should include information on the source, background of fishes
- line 119 change to "alternatives to antibiotic techniques"
- lines 220 - 230 lots of spacing errors
- line 261 headers should be on new lines
- line 275 indentation error
- line 281 punctuation errors
- lines 332-333 spacing and formatting error
- lines 348-353 left justify
- lines 440, 443 unclear; not sentences
- References not formatted correctly. Species names not formatted correctly. Some article titles in caps, others in lowercase (be consistent). Lines 507, 519, 558 spacing errors. 549 incomplete citation.
- I can't read Table 1
- lines 619, 624 spacing errors
- figure 6 very few of these phyla are visible here

Reviewer #3 (Comments for the Author):

The study is generally appropriate, but the authors need to have appropriate answers to the following questions:

- The stages of laboratory evaluations in the materials and methods section do not match the results section. Lactobacilli should be screened first and then the antimicrobial activity should be checked. LLK48 is introduced before the screening.
- In a separate group, uninfected fish could be evaluated with probiotic bacteria as controls.
- In a separate group, a pretreatment with probiotic bacteria could be evaluated for its influence on the onset and severity of the infection.
- On what basis was the number of probiotic inoculated bacteria selected?
- Why have growth performance and immunomodulatory function not been evaluated?

Staff Comments:

Preparing Revision Guidelines

Please return the manuscript within 60 days; if you cannot complete the modification within this time period, please contact me. If you do not wish to modify the manuscript and prefer to submit it to another journal, please notify me of your decision immediately so that the manuscript may be formally withdrawn from consideration by Microbiology Spectrum.

Date Due: May 09, 2022

Manuscript #Spectrum 01128-22

Title: *Lactococcus lactis* treatment of *Streptococcus agalactiae* infections affecting the structure of intestinal microbiota in zebrafish (*Danio rerio*)

In all manuscript there are several mistakes in grammar and sentence construction. The authors must check it carefully.

Overall Recommendation:

The authors isolated a strain of *Lactococcus lactis* KUST48 from tilapia with *in vitro* antimicrobial activity against *Streptococcus agalactiae*. They studied the effect of *L. lactis* strain on zebrafish infected with *S. agalactiae* from the perspective of the intestinal microbiota. The *in vivo* experimental test was divided into three groups: the control group (A) injected with PBS, the infected group (B) injected with *S. agalactiae*, and the treated group (C) injected with *S. agalactiae* and 24h post-injection with *L. lactis* KUST48. Then, the 16S rRNA gene sequences of the intestinal microbiota of these three groups were sequenced using Illumina high-throughput sequencing technology.

The topic of this study is very interesting. The results of this study showed that the injection of *L. lactis* KUST48 produced benefits on zebrafish infected with *S. agalactiae*. However, the authors refer to *L. lactis* as a probiotic when this was not yet demonstrated (see Abstract review). Moreover, the term “treatment” referred in the Title would not be appropriate since a more comprehensive evaluation is needed, where different doses, times, and effects of *L. lactis* on infected zebrafish are tested or evaluated. On the other hand, the manuscript was hard to read, the authors should find a native speaker to polish the English.

For a better understanding of the effect of *L. lactis* KUST48 on the zebrafish intestinal microbiota, another control group with the *L. lactis* KUST48 and without the *S. agalactiae* would be useful to add.

Title:

- I suggest the following title: *Lactococcus lactis* effect on the intestinal microbiota of *Streptococcus agalactiae* infected zebrafish

Abstract:

- The abstract is very poorly written and needs to be rewritten.
- Lines 39-41. In importance section “...The probiotic *Lactococcus lactis* (LLK48) can be considered as a therapeutic and water quality modifying agent for *Streptococcus agalactiae* (SA) infections ...”.

- It is not appropriate to speak of "probiotic" (throughout the manuscript), because both, other tests are needed to confirm that a bacterial strain can be considered a probiotic (i.e. antibiotic resistance genes, haemolytic activity, bile tolerance, etc.), and the intraperitoneal injection is not the proper administration way. A probiotic is administered with the food or in the rearing water. Thus, they should change "probiotic" by "potential probiotic".
- In the present study, the effect of *L. lactis* (LLK48) on water quality was not demonstrated nor evaluated. Please check this.

Keywords: Use different keywords that do not present in the title to facilitate searching on your paper

Introduction

- The Introduction is very poorly written and needs to be rewritten. Some examples: lines 46-49; 77-78; 81-87; 88-91; etc.
- Line 60, the bibliographic citation (6) is not adequate. Please use a proper citation.
- Lines 64-66. Bibliographic citation (9) does not support the written sentence.
- Lines 60-63. The bibliographic citation is not adequate. There are several papers about of the beneficial uses of probiotics in Aquaculture. You must use an appropriate citation.
- Lines 74-76: this sentence is confusing; please indicate if these data are being published in another journal or are part of this work.
- Lines 108-118. This information should be in materials and methods section.

Materials and methods:

Section 2.1

- Please change the title: "*Isolation of intestinal flora*" by "*LAB isolation from tilapia intestinal tract*".
- Please add the proportion of the dilutions performed for the isolation assay.
-

Section 2.2: in this section there are several mistakes. Please check it carefully

- I suggest changing the title: "*Screening of antibacterial strains*" by "*Screening of strains with antimicrobial activity*".
- Please add the origin and culture conditions of the *S. agalactiae* strain.

- Please add the dilutions performed for the minimum inhibitory concentration (MIC) assay.
- Line 141: please clarify if it is broth or semi-solid medium, if it is semi-solid medium, indicate the agar concentration.
- Lines 142: does the plate really invert to mix?
- Line 145: Please add Kanamycin concentration.

Section 2.3:

- Line 152-153: Please review the content of the following sentence: “*In order to select a lactic acid bacterial strain with good antimicrobial activity against S. agalactiae, total DNA was extracted using a DNA Extraction Kit...*”
- Line 155: “*...of 16S gene sequencing...*”, please replace with “*...of 16S rRNA gene sequencing...*”
- Line: 165-167: Please review the following sentence: “*The sequenced 16S was searched in GenBank of the National Center for Biotechnology Information (NCBI) (<https://www.ncbi.nlm.nih.gov/>).*”

Section 2.4:

- I suggest changing the title for “Zebrafish and experimental design”
- Lines 178-181: The content of these lines should be in section 2.2.
- Line 192: “*Thirty zebrafish in each group were then randomly divided into three special aquarium divider box...*”. Please explain which groups you are referring to.

Section 2.5

- In lines 210-212 the text is confusing: “...the intestines of 15 zebrafish were randomly divided into 5 portions, and each portion of intestine was mixed into one sequencing sample for each group”. Please rewrite this sentence and explain better how each sample was prepared.
- In materials and methods, please indicate the period of time used to calculate the mortality rate

In materials and methods, please add a new section that details the prediction method of the functional changes of the intestinal microbial community (PICRUSt).

Section 2.8

- Please add the statistical analysis of the predicted functions using PICRUSt.

Results

Section 3.1:

- Please change the title: “*Isolation and bacteriostatic experiment of intestinal bacteria*” by “*Isolation and antimicrobial activity of intestinal bacteria*”.
- Throughout the manuscript, in the objective of this study (lines 103-104), in section 2.1 of materials and methods and in the title of section 3.1 of results, several isolated bacteria are mentioned (several isolates belonging to the group of LAB), however the result of only one is showed. I suggest adding a table with the results of antimicrobial activity against *S. agalactiae* of the 79 LAB strains isolated from tilapia intestine. Please add also the results of the minimum inhibitory concentration (MIC) described in section 2.2 of materials and methods.
- In figure legends, for Figure 1, it would be appropriate to change the name of the strain "LLK48" to "strain KUST48" since at this point the strain has not yet been identified.
- Line 269: there are several mistakes in grammar and sentence construction. Please check it carefully. Moreover, please change “...*the 16S rDNA sequence*...” by “...*16S rRNA sequence*...”

Section 3.2:

- In this section (lines 276-287) there are several mistakes in grammar and sentence construction. Please check it carefully.

Section 3.3

- Please change the title: “*Optimization analysis of high-throughput sequencing data*” by “*Analysis of sequences*”.

Section 3.4

- Lines 306-307. Please, check “Actinobacteria and Fusobacteria” are repeated.
- There are several grammatical mistakes in this paragraph. Please check it carefully

Section 3.5

- Lines 316. “*Escherichia-Shigella* and *Aeromonas* were the main genera (the sum of the relative abundance > 0.1%) in all the three groups (Table 2)”. Please check, in table 2 the *Aeromonas* genus does not appear.

- There are several mistakes in grammar and sentence construction. Please check it carefully.

Section 3.6

- Please check Table 3, the information mentioned in lines 355-359 does not appear.
- Lines 360-366 are repeated.

Section 3.7

- Lines 369-370. “*The analysis results of species relative abundance in each group at the genus level are shown in Figure 5*”. Please check this sentence, in Figure 5 the analysis of microbial community composition at the phylum level is showed.
- The results in the section 3.7 are similar to that shown in the section 3.5, please unify the results in a single section.
- The results are very poorly presented. The manuscript has many grammatical mistakes, making it very difficult to read.
- It would be interesting to analyze the relative abundance of specific bacterial genera such as *Vibrio* sp., *Lactococcus* sp., for the different groups

Section 3.8

- Please indicate the significant differences in predicted microbial functions among the three groups. This section was poorly written. Please rewrite it.

Table 2:

- “*Different superscripts identify significant differences ($P < 0.05$) in relative abundance*”. Please indicate whether the significant differences are within the same group or between groups.

Table 3:

- Please indicate the meaning of the categories (A, B, C, etc.) and the values that appear in Table 3. Indicate the results of the statistical analysis among 3 groups.

Figure legends

- Fig. 1: Please, change “Bacteriostatic effect of *LLK48* on *S. agalactiae*. (a) Positive control: Antibacterial effect of Kanamycin on *S. agalactiae*; (b) Bacteriostatic effect of *LLK48* on *S. agalactiae*; (c) Bacteriostatic effect of sterile distilled water on *S.*

agalactiae.”; by “Bacteriostatic effect of KUST48 strain on *S. agalactiae*. (a) Positive control: Kanamycin; (b) KUST48 supernatant; (c) Negative control: sterile distilled water.”

- Fig 3: Please, change the figure title and expand the description. The title may be: *Shannon index analysis of the zebrafish intestinal samples*. Add details to the legend of figure 3. Moreover, it remains to explain the meaning of a, b and c
- Fig 4: Please, change the figure title, it may be: “Principal component analysis (PCA) of the zebrafish intestinal samples”. Add the meaning of a, b and c.
- Fig 5: Please, change the figure title, it may be: “Analysis of microbial community composition at the phylum level in zebrafish intestinal samples”. Describe the meaning of a1, a2, a3, ... etc
- Fig 6: This figure provides the same information as figure 5, I suggest removing it.

Discussion

- Lines 421-422: Please cite articles related to aquaculture probiotics

For Reviewer #1:

Authors have well revised according to the comments.

Response: *We sincerely appreciated the reviewer's review and positive comments. we carefully revised and responded the reviewers' comments. Please see point-by-point response as follow.*

1. the term "treatment" referred in the Title would not be appropriate, I suggest the following title: *Lactococcus lactis* effect on the intestinal microbiota of *Streptococcus agalactiae* infected zebrafish.

Response: *Thank for the reviewer's professional review. We have changed the title to *Lactococcus lactis* effect on the intestinal microbiota of *Streptococcus agalactiae* infected zebrafish (*Danio rerio*). Please see **lines 1-2**.*

2. the manuscript was hard to read, the authors should find a native speaker to polish the English

Response: *the whole manuscript was submitted to language editing service by native English in **Charles Lois Edit** (<https://editingservices.wiley.cn/>), as suggested by the Reviewers. A certification is provided in the point-by-point response. the proof of touch-up is uploaded to the system in the form of a file.*

3. For a better understanding of the effect of *L. lactis* KUST48 on the zebrafish intestinal microbiota, another control group with the *L. lactis* KUST48 and without the *S. agalactiae* would be useful to add.

Response: *Thanks to the reviewers for their valuable comments. First, referring to the grouping method used in this experiment (Shan, 2021), DOI: doi: 10.1016/j.lwt.2020.110826, we mainly explored the response mechanism of LLK48 to the gut of zebrafish infected with *Streptococcus agalactiae*. Second, the experiments involved in this paper have been completed. The experimental conditions described in this paper and the newly added LLK48 group will have different effects on zebrafish in the group with late addition of *Lactococcus lactis* LLK48 without *S. agalactiae*. Therefore, we did not add this part. I hope the reviewers understand our difficulties. However, we are very concerned about the issues raised by the reviewers. We used a new batch of zebrafish to test the effect of the probiotic LLK48 on zebrafish. The results showed that the results of adding LLK48 were similar to the PBS group.*

4. The abstract is very poorly written and needs to be rewritten.

Response: *We have revised the ABSTRACT, as suggested.*

5. Lines 39-41. In importance section "...The probiotic *Lactococcus lactis* (LLK48) can be considered as a therapeutic and water quality modifying agent for *Streptococcus*

agalactiae (SA) infections ...".- It is not appropriate to speak of "probiotic" (throughout the manuscript), because both, other tests are needed to confirm that a bacterial strain can be considered a probiotic (i.e. antibiotic resistance genes, haemolytic activity, bile tolerance, etc.), and the intraperitoneal injection is not the proper administration way. A probiotic is administered with the food or in the rearing water. Thus, they should change "probiotic" by "potential probiotic".

Response: Thank for the reviewer's reminder. We have changed probiotics to potential probiotics. Please *see lines 38*.

6. In the present study, the effect of *L. lactis* (LLK48) on water quality was not demonstrated nor evaluated. Please check this.

Response: The effect of *L. lactis* (LLK48) on water quality was not demonstrated nor evaluated. Therefore, we have revised the important part of the article. Please *see lines 38-39*

7. Keywords: Use different keywords that do not present in the title to facilitate searching on your paper

Response: We have modified the keywords according to the reviewer's comments. Please *see lines 42*.

8. The Introduction is very poorly written and needs to be rewritten. Some examples: lines 46-49;

Response: We have rephrased the sentence in the revised manuscript, as suggested. Please *see lines 44-47*.

9. Line 60, the bibliographic citation (6) is not adequate. Please use a proper citation.

Response: According to the reviewers' opinions, references have been supplemented to explain. Please *see lines 58*.

10. Lines 60-63. The bibliographic citation is not adequate. There are several papers about the beneficial uses of probiotics in Aquaculture. You must use an appropriate citation.

Response: References on the application of probiotics in aquaculture were revised, as suggested. Please *see lines 60*.

11. Lines 64-66. Bibliographic citation (9) does not support the written sentence.

Response: Added reference supplement. Please *see lines 63-64*.

12. Lines 74-76: this sentence is confusing; please indicate if these data are being published in another journal or are part of this work.

Response: These data have not been published, and we have revised it, as suggested. Because the order of the article has been adjusted, please *see lines 285*.

13. The Introduction is very poorly written and needs to be rewritten. Some

examples:77-78; 81-87; 88-91; etc.

Response: We have rewritten, as suggested. Please see lines73-74,76-82,93-96.

14. Lines 108-118. This information should be in materials and methods section

Response: This content has been placed in materials and methods. Please see lines 239-242,260-262.

15. Please change the title: "Isolation of intestinal flora" by "LAB isolation from tilapia intestinal tract.

Response: As suggested, we have changed the title to LAB isolation from tilapia intestinal tract. Please see lines 105.

16.Please add the proportion of the dilutions performed for the isolation assay

Response: Dilution ratio of added separation test. Please see lines 112-118.

17.I suggest changing the title: "Screening of antibacterial strains" by "Screening of strains with antimicrobial activity".

Response: The title of 2.2 has been changed to "Screening of strains with antimicrobial activity". Please see lines 120.

18. Please add the origin and culture conditions of the *S. agalactiae* strain.

Response: Added the origin and culture conditions of the *S. agalactiae* strain, as suggested. Please see lines 124-128.

19.Please add the dilutions performed for the minimum inhibitory concentration (MIC) assay.

Response: According to the reviewer's suggestions, we added this part. Please see lines 144-162.

20.Line 141: please clarify if it is broth or semi-solid medium, if it is semi-solid medium, indicate the agar concentration.

Response: After inspection, we determined that it was semi-solid medium, and the agar concentration was 0.75%. Please see lines 134-135.

21.Lines 142: does the plate really invert to mix?

Response: Shake gently and pour into the plate. Please see lines135.

22.Line 145: Please add Kanamycin concentration

Response: Added Kanamycin concentration. Please see lines138.

23.Line 152-153: Please review the content of the following sentence: "In order to select a lactic acid bacterial strain with good antimicrobial activity against *S. agalactiae*, total DNA was extracted using a DNA Extraction Kit..."

Response: Checked and changed to identify lactic acid strains with good

antibacterial activity against *S. agalactiae*, total DNA was extracted using a DNA extraction kit. Please *see lines 164-165*.

24. Line 155: "...of 16S gene sequencing...", please replace with "...of 16S rRNA gene sequencing..."

Response: Changed to of 16S rRNA gene sequencing. Please *see lines 168*.

25. Line: 165-167: Please review the following sentence: "The sequenced 16S was searched in GenBank of the National Center for Biotechnology Information (NCBI) (<https://www.ncbi.nlm.nih.gov/>)."

Response: Checked, Please *see lines 177-178*.

26. I suggest changing the title for "Zebrafish and experimental design".

Response: Changed title to "Zebrafish and experimental design". Please *see lines 182*.

27. Lines 178-181: The content of these lines should be in section 2.2.

Response: Content adjusted to 2.2, Please *see lines 124-126*.

28. Line 192: "Thirty zebrafish in each group were then randomly divided into three special aquarium divider box...". Please explain which groups you are referring to.

Response: The control group (a) injected with PBS, the infected group (b) injected with *S. agalactiae*, and the treated group (c) injected with *S. agalactiae*. Please *see lines 200*.

29. In lines 210-212 the text is confusing: "...the intestines of 15 zebrafish were randomly divided into 5 portions, and each portion of intestine was mixed into one sequencing sample for each group. Please rewrite this sentence and explain better how each sample was prepared.

Response: Modified according to comments. Please *see lines 215-220*.

30. In materials and methods, please indicate the period of time used to calculate the mortality rate.

Response: The death was observed every 24h. The time for calculating mortality was 120 h. Please *see lines 211-212*.

31. In materials and methods, please add a new section that details the prediction method of the functional changes of the intestinal microbial community (PICRUSt).

Response: This section has been added in materials and methods. Please *see lines 270-281*.

32. Section 2.8, Please add the statistical analysis of the predicted functions using PICRUSt.

Response: This section has been added in materials and methods. Please *see lines 278-281*.

33 Please change the title: "Isolation and bacteriostatic experiment of intestinal bacteria" by "Isolation and antimicrobial activity of intestinal bacteria".

Response: *Changed to Isolation and antimicrobial activity of intestinal bacteria. Please see lines 283.*

34. Throughout the manuscript, in the objective of this study (lines 103-104), in section 2.1 of materials and methods and in the title of section 3.1 of results, several isolated bacteria are mentioned (several isolates belonging to the group of LAB), however the result of only one is showed. I suggest adding a table with the results of antimicrobial activity against *S. agalactiae* of the 79 LAB strains isolated from tilapia intestine. Please add also the results of the minimum

Response: *as suggested The data has been added. Please see Table 1.*

35. Please add also the results of the minimum inhibitory concentration (MIC) described in section 2.2 of materials and methods

Response: *MIC results have added according to the reviewer's suggestions. Please see lines 289.*

36. In figure legends, for Figure 1, it would be appropriate to change the name of the strain "LLK48" to "strain KUST48" since at this point the strain has not yet been identified.

Response: *For Figure 1, the name of strain "LLK48" has been changed to "Strain KUST48". See Figure 1*

37. Line 269: there are several mistakes in grammar and sentence construction. Please check it carefully. Moreover, please change "...the 16S rDNA sequence..." by "...16S rRNA sequence.

Response: *Changed to 16S rRNA sequence. Please see lines 290.*

38. Section 3.2 In this section (lines 276-287) there are several mistakes in grammar and sentence construction. Please check it carefully.

Response: *Modified as required, please see lines 296-307.*

39. Section 3.3 Please change the title: "Optimization analysis of high-throughput sequencing data" by "Analysis of sequences.

Response: *Changed to Analysis of sequences. Please see lines 308.*

40. Lines 306-307. Please, check "Actinobacteria and Fusobacteria" are repeated

Response: *Thank you very much for your careful review. Checked, deleted duplicate Checked, deleted duplicate. Please see lines 325-326.*

41. Section 3.4 There are several grammatical mistakes in this paragraph. Please check it carefully

Response: It has been revised as required by the reviewe. Please see Section 3.4

42.Lines 316. "Escherichia-Shigella and Aeromonas were the main genera (the sum of the relative abundance > 0.1%) in all the three groups (Table 2)". Please check, in table 2 the Aeromonas genus does not appear.

Response: Thanks for the reviewer's reminder. Due to our negligence, the information is wrong, and we have made changes. Please see lines 335.

43.Section 3.5 There are several mistakes in grammar and sentence construction. Please check it carefully.

Response: It has been revised as required by the reviewer Please see Section 3.5

44.Please check Table 3, the information mentioned in lines 355-359 does not appear.

Response:Thanks for the reviewer's reminder. Due to our negligence, the information is wrong, and we have made changes.This information can be found in Table 4.We have also made corresponding explanations in the article

45.Lines 360-366 are repeated.

Response: Duplicate rows deleted.

46. The results in the section 3.7 are similar to that shown in the section 3.5, please unify the results in a single section. The results are very poorly presented. The manuscript has many grammatical mistakes, making it very difficult to read

Response: The contents of 3.7 have been unified into 3.5 Please see lines 341-348.

47. Lines 369-370. "The analysis results of species relative abundance in each group at the genus level are shown in Figure 5". Please check this sentence, in Figure 5 the analysis of microbial community composition at the phylum level is showed.

Response:Thanks for the comments made by the reviewers. Due to our negligence, a misunderstanding has been caused. We modified and moved section 3.7 to section 3.5

48.-It would be interesting to analyze the relative abundance of specific bacterial genera such as Vibrio sp., Lactococcus sp., for the different groups

Response: Figure 3 shows bacteria with high abundance and low abundance of Lactobacillus, suggesting a competitive role between LLK48 and S. agalactiae in this experiment.

49.Section 3.8

Please indicate the significant differences in predicted microbial functions among the thre groups. This section was poorly written. Please rewrite it.

Response: We have check it, and revised, as suggested. Please see lines 375-383.

50.Table 2,"Different superscripts identify significant differences ($P < 0.05$) in relative abundance". Please indicate whether the significant differences are within the same

group or between groups.

Response: According to the reviewers' opinions, we intend to make changes. Please see table 3

51. Table 3:- Please indicate the meaning of the categories (A, B, C, etc.) and the values that appear in Table 3. Indicate the results of the statistical analysis among 3 groups.

Response: According to the reviewer's opinion, the meaning of a, b and c has been added. Please see table 5

52.-Fig. 1: Please, change "Bacteriostatic effect of *LLK48* on *S. agalactiae*. (a) Positive control: Antibacterial effect of Kanamycin on *S. agalactiae*; (b) Bacteriostatic effect of *LLK48* on *S. agalactiae*; (c) Bacteriostatic effect of sterile distilled water on *S. agalactiae*."; by "Bacteriostatic effect of *KUST48* strain on *S. agalactiae*. (a) Positive control: Kanamycin; (b) *KUST48* supernatant; (c) Negative control: sterile distilled water."

Response: We have modified the drawing as required, Please see FIG 1.

53. Fig 3: Please, change the figure title and expand the description. The title may be: Shannon index analysis of the zebrafish intestinal samples. Add details to the legend of figure 3. Moreover, it remains to explain the meaning of a, b and c.

Response: Revised according to the reviewer's comments. Please see FIG 3.

54. Fig 4: Please, change the figure title, it may be: "Principal component analysis (PCA) of the zebrafish intestinal samples". Add the meaning of a, b and c.

Response: Revised according to the reviewer's comments. Please see FIG 4.

55. Fig 5: Please, change the figure title, it may be: "Analysis of microbial community composition at the phylum level in zebrafish intestinal samples". Describe the meaning of a1, a2, a3, ... etc

Response: Revised according to the reviewer's comments. Please see FIG 5.

56. Fig 6: This figure provides the same information as figure 5, I suggest removing it.

Response: Revised according to the reviewer's comments, Delete Figure 6

57. Discussion

- Lines 421-422: Please cite articles related to aquaculture probiotics.

Response: References have been added as required, Please see lines 414-417.

Reviewer #2

1. lines 2,3 species name is not properly formatted

Response: *The name of the species has been modified as suggested. Please see lines 1-2.*

2. lines 8,9 addresses should be on separate lines

Response: *Thanks for the reviewer's reminder, It has been modified according to the reviewer's requirements. Please see lines 8-9.*

3. line 33 "tilapia" should not be capitalized (it is not a proper noun)

Response: *It has been changed to the correct format according to the reviewer's comments. Please see lines 34.*

4. line 42 ineffective keywords

Response: *Keywords replaced, Please see lines 42.*

5. line 83 replace indexes with indices

Response: *Replaced with indices, Please see lines 78.*

6. lines 87 88, indentation is inconsistent

Response: *We have adjusted the indent as required, Please see lines 93-94.*

7. line 99 correct species nomenclature

Response: *Changed to correct species name, Please see lines 90.*

8. line 97 methods should include information on the source, background of fishes

Response: *According to the reviewer's comments, Please see lines 183-190.*

9. line 119 change to "alternatives to antibiotic techniques"

Response: *changed to "alternatives to antibiotic techniques" Please see lines 103.*

10. lines 220 - 230 lots of spacing errors

Response: *Thanks for your comments, we have adjusted the spacing. Please see lines 222-237.*

11. line 261 headers should be on new lines

Response: *Thanks for the reviewer's reminder. The title has been changed to a separate line. Please see lines 282.*

12. line 275 indentation error

Response: *Thanks for the reviewer's reminder, Modified to correct, Please see lines 296-307.*

13. line 281 punctuation errors

Response: Thanks for the reviewer's reminder. The redundant punctuation has been deleted. Please see lines 302.

14.lines 332-333 spacing and formatting error

Response: The line spacing has been adjusted according to the reviewer's comments. Please see lines 354-355.

15.lines 348-353 left justify

Response: Left aligned according to reviewer's comments. Please see lines 367-373.

16.lines 440, 443 unclear; not sentences

Response: Please see lines 433,435-437.

17.References not formatted correctly. Species names not formatted correctly. Some article titles incaps, others in lowercase (be consistent). Lines 507, 519, 558 spacing errors. 549 incompletecitation.

Response: As suggested by the reviewers, we have adjusted the line spacing throughout the article and introduced new literature for clarification. Please see lines 573.

18.I can't read Table 1

Response: Table 1 showed the adequate sequencing of 15 samples. Now, We adjusted the table, Please see Table 2. and explain Table 1.

19.lines 619, 624 spacing errors

Response: Because Figure 6 was deleted according to the requirements of reviewer 1, the information of Figure 6 in line 619 was also deleted, However, we have adjusted the line spacing throughout the article

20.figure 6 very few of these phyla are visible here

Response: Since Figure 6 and figure 5 express the same meaning, Figure 6 is deleted according to the opinion of reviewer 1, and the phyla level can be checked in Figure 5

Reviewer #3

1. The stages of laboratory evaluations in the materials and methods section do not match the results section. Lactobacilli should be screened first and then the antimicrobial activity should be checked. LLK48 is introduced before the screening.

Response: As the reviewers reminded us, we matched the materials to the methods and results and adapted the content to some extent. Introducing LLK48 before screening, Please see *lines 106*.

2. In a separate group, uninfected fish could be evaluated with probiotic bacteria as controls.

Response: Thanks for the professional comments of the reviewers. We attached great importance to the questions raised by the reviewers. We re conducted the experiment and used probiotics to evaluate the non infected zebrafish. The results showed that the results of probiotics on the non infected zebrafish group and PBS group were similar. However, as the experiment involved in the article has been completed, the probiotics on the non infected zebrafish control group and the experiment in the article, We cannot guarantee the consistency of the zebrafish and experimental conditions. Therefore, probiotics will not be included in the article for uninfected ones. I hope reviewers can make a approval for the responses.

3. In a separate group, a pretreatment with probiotic bacteria could be evaluated for its influence on the onset and severity of the infemction.

Response: Thanks to the reviewer's valuable comments. The experiment has completed. As you known, if we want to add probiotics to the effect of zebrafish, the experimental conditions involved in the article and the effect of probiotics on zebrafish will be different. Therefore, this part of the content was not added, and I hope the reviewers can understand our difficulties. However, we are very concerned about the questions raised by the reviewers. We used a new batch of zebrafish to conduct experiments on the effect of probiotics on zebrafish, and we did not observe the phenomenon of infection.

4. On what basis was the number of probiotic inoculated bacteria selected?

Response: A 50% survival rate was reported for zebrafish at 24h post-injection, indicating that a 10^6 CFU/mL concentration of *S. agalactiae* could successfully induce the immune response in the whole body of adult zebrafish before 24 hpi (Patterson H, 2012). doi:10.1016/j.dci.2012.07.007. According to previous experimental studies in our laboratory, the half lethal concentration of *S. agalactiae* for zebrafish was 10^6 CFU/mL (Zhang et al., 2019). doi:10.2848.10.3389/fmicb.2019.02848. Thus, *S. agalactiae* concentration of 10^6 CFU/mL was employed to trigger the immune response of zebrafish in downstream experiments.

5. Why have growth performance and immunomodulatory function not been evaluated?

Response: In our experiment, the results showed that there was no significant

difference in the growth performance of the three groups of zebrafish. For the evaluation of immune performance mentioned by the reviewer, based on the previous research in the laboratory, there has been a comprehensive evaluation of the immune performance of zebrafish (Luo et al., 2021). doi:10.1016/j.aqrep.2021.100639. In addition, this experiment focuses on exploration of the intestinal response mechanism of Lactococcus lactis to zebrafish infected with Streptococcus lactis. Therefore, the immune performance of zebrafish in this experiment has not been evaluated.

Others:

the whole manuscript was submitted to language editing service by native English in

Charles Lois Edit (<https://editingservices.wiley.cn/>), as suggested by the Reviewers.

A certification is provided in *the point-by-point response*. The proof of touch-up is uploaded to the system in the form of a file.

For the entire article, we have made some adjustments to the content to move and delete, so that reviewers can better understand our article

lines 84-86: We have rephrased this sentence to make the article smoother.

lines 67: Because the description is unclear, we have added the genus name.

lines 108: Supplementary operation of tilapia euthanasia

lines 110: Add saline concentration

lines 114-115: Add instrument manufacturer, Shanghai Yiheng

lines 187: Add zebrafish purchase Internet site.

lines 366: Changed 3.6 title to Diversity Analysis.

August 12, 2022

Dr. Chunyan Tan
Kunming University of Science and Technology
Yunnan , Kunming
China

Re: Spectrum01128-22R1 (Lactococcus lactis effect on the intestinal microbiota of Streptococcus agalactiae infected zebrafish (Danio rerio))

Dear Dr. Chunyan Tan:

Link Not Available

Sincerely,

Konstantinos Kormas

Journals Department
Reviewer comments:

Reviewer #1 (Comments for the Author):

Keywords: Use different keywords that do not present in the title to facilitate searching on your paper

Overall Recommendation

The authors should unify the terminology to define the experimental groups, and use them throughout the manuscript, because they use different definitions throughout the manuscript.

In the abstract:

"This study divided zebrafish into three groups: control group, injected with phosphate-buffered saline; infected group, injected

with *S. agalactiae*; and treatment group, treated with LLK48 after *S. agalactiae* injection"

In section 2.7:

"...in negative control groups, *S. agalactiae*-infected groups (*S. agalactiae* control), and LLK48-treated groups (treatment)"

In section 2.4:

"Thirty zebrafish in each group (control group, infection group, and treatment group)"

In section 2.8:

"...PBS control-*S. agalactiae* control-LLK48 treatment"

Materials and methods:

Section 2.2: Please check it carefully

- Line 141: I suggest changing: "The diameter of the bacteriostatic circle indicates the bacteriostatic activity of LLK48 on *S. agalactiae*" by "The diameter of the inhibition zone indicates the bacteriostatic activity of the LAB supernatants on *S. agalactiae*".

- Line 144-146: "To determine the minimum inhibitory concentration (MIC), the concentration of supernatant was detected using the Pierce® BCA Protein Assay Kit ..." Please check it carefully. One does not determine the concentration of a supernatant, in any case the protein concentration, of a metabolite, of bacteria, etc.

To quantify the amount of antimicrobial activity of a bacterium and determine the minimum inhibitory concentration (MIC) I recommend the following paper:

Sequeiros, C., Garcés, M. E., Vallejo, M., Marguet, E. R., & Olivera, N. L. (2015). Potential aquaculture probiont *Lactococcus lactis* TW34 produces nisin Z and inhibits the fish pathogen *Lactococcus garvieae*. *Archives of microbiology*, 197(3), 449-458. Idem, Line 161

Section 2.8

- Line 261: Please check it carefully: "We analyzed 16S rRNA amplicons to generate changes in the diversity".

Results

Section 3.1:

- Table 1: The diameter of the inhibition zone of the negative control appears as 8 mm. Please check it carefully.

Reviewer #2 (Comments for the Author):

I am favorably impressed with the novelty and character of this work. The initial review was taken constructively and my own recommendations have all been followed responsibly and professionally. I am confident about the professionalism and integrity displayed in the processing of this manuscript.

Staff Comments:

Preparing Revision Guidelines

Please return the manuscript within 60 days; if you cannot complete the modification within this time period, please contact me. If you do not wish to modify the manuscript and prefer to submit it to another journal, please notify me of your decision immediately so that the manuscript may be formally withdrawn from consideration by Microbiology Spectrum.

If your manuscript is accepted for publication, you will be contacted separately about payment when the proofs are issued;

please follow the instructions in that e-mail. Arrangements for payment must be made before your article is published. For a complete list of **Publication Fees**, including supplemental material costs, please visit our website.

Date Due: Aug 01, 2022

Manuscript #Spectrum 01128-22

Title: *Lactococcus lactis* effect on the intestinal microbiota of *Streptococcus agalactiae* infected zebrafish (*Danio rerio*)

Keywords: Use different keywords that do not present in the title to facilitate searching on your paper

Overall Recommendation

The authors should unify the terminology to define the experimental groups, and use them throughout the manuscript, because they use different definitions throughout the manuscript.

In the abstract:

“This study divided zebrafish into three groups: control group, injected with phosphate-buffered saline; infected group, injected with *S. agalactiae*; and treatment group, treated with LLK48 after *S. agalactiae* injection”

In section 2.7:

“...in negative control groups, *S. agalactiae*-infected groups (*S. agalactiae* control), and LLK48-treated groups (treatment)”

In section 2.4:

“Thirty zebrafish in each group (control group, infection group, and treatment group)”

In section 2.8:

“...PBS control-*S. agalactiae* control-LLK48 treatment”

Materials and methods:

Section 2.2: Please check it carefully

- Line 141: I suggest changing: “The diameter of the bacteriostatic circle indicates the bacteriostatic activity of LLK48 on *S. agalactiae*” by “The diameter of the inhibition zone indicates the bacteriostatic activity of the LAB supernatants on *S. agalactiae*”.

- Line 144-146: “To determine the minimum inhibitory concentration (MIC), the concentration of supernatant was detected using the Pierce® BCA Protein Assay Kit ...” Please check it carefully. One does not determine the concentration of a supernatant, in any case the protein concentration, of a metabolite, of bacteria, etc.

To quantify the amount of antimicrobial activity of a bacterium and determine the minimum inhibitory concentration (MIC) I recommend the following paper:

Sequeiros, C., Garcés, M. E., Vallejo, M., Marguet, E. R., & Olivera, N. L. (2015). Potential aquaculture probiont *Lactococcus lactis* TW34 produces nisin Z and inhibits the fish pathogen *Lactococcus garvieae*. Archives of microbiology, 197(3), 449-458.

Idem, Line 161

Section 2.8

- Line 261: Please check it carefully: “We analyzed 16S rRNA amplicons to generate changes in the diversity”.

Results

Section 3.1:

- **Table 1:** The diameter of the inhibition zone of the negative control appears as 8 mm. Please check it carefully.

-

For Reviewer #1:

1. Keywords: Use different keywords that do not present in the title to facilitate searching on your paper.

Response: Thank you to the reviewers for their professional review. We have made changes as suggested, please see line 42-43.

2. The authors should unify the terminology to define the experimental groups, and use them throughout the manuscript, because they use different definitions throughout the manuscript.

Response: Thank the reviewers for their professional review and patient guidance, We have used uniform terminology to define the experimental group, such as control group, infection group, and treatment group. Please check.

3. In the abstract:

"This study divided zebrafish into three groups: control group, injected with phosphate-buffered saline; infected group, injected with *S. agalactiae*; and treatment group, treated with LLK48 after *S. agalactiae* injection"

Response: In the abstract, We have named the group control group, infection group, and treatment group. please see line 17-19.

4. In section 2.4:

"Thirty zebrafish in each group (control group, infection group, and treatment group)"

Response: In the section 2.4, We changed it into control group, infection group, and treatment group. please see line 194.

5. In section 2.7:

"...in negative control groups, *S. agalactiae*-infected groups (*S. agalactiae* control), and LLK48-treated groups (treatment)"

Response: In the section 2.7, We changed it into control group, infection group, and treatment group. please see line 235-236.

6. In section 2.8:

"...PBS control-*S. agalactiae* control-LLK48 treatment"

Response: In the section 2.8, We changed it into control group, infection group, and treatment group. please see line 255.

7. Section 2.2: Please check it carefully

- Line 141: I suggest changing: "The diameter of the bacteriostatic circle indicates the bacteriostatic activity of LLK48 on *S. agalactiae*" by "The diameter of the inhibition zone indicates the bacteriostatic activity of the LAB supernatants on *S. agalactiae*".

Response: Thank the reviewers for their professional review and patient guidance. We have revised it as suggested, please see line 142-143.

8. Line 144-146: "To determine the minimum inhibitory concentration (MIC), the concentration of supernatant was detected using the Pierce® BCA Protein Assay Kit ..." Please check it carefully. One does not determine the concentration of a supernatant, in any case the protein concentration, of a metabolite, of bacteria, etc. To quantify the amount of antimicrobial activity of a bacterium and determine the minimum inhibitory concentration (MIC) I recommend the following paper: Sequeiros, C., Garcés, M. E., Vallejo, M., Marguet, E. R., & Olivera, N. L. (2015). Potential aquaculture probiont *Lactococcus lactis* TW34 produces nisin Z and inhibits the fish pathogen *Lactococcus garvieae*. Archives of microbiology, 197(3), 449-458. Idem, Line 161.

Response: Thank you to the reviewers for their professional review, We removed the reference to the Pierce® BCA Protein Assay Kit to detect the concentration of the supernatant. We also carefully studied the reference papers given by the reviewers and re-performed the MIC measurement experiments with the laboratory conditions. please see line 146-156. And the corresponding results are obtained, please see line 291. In addition, please see the following table for supplementary explanation of this experiment

CFS concentration mg/mL	1000	500	250	125	62.5	31.25	15.625	7.8125
Changes in OD600	-	-	-	-	+	+	+	+

"-": No significant difference in the control group, "+": Have significant difference in the control group.

9. Section 2.8

• Line 261: Please check it carefully: "We analyzed 16S rRNA amplicons to generate changes in the diversity".

Response: We have rephrased the sentence in the revised manuscript, as suggested. Please see lines 254-256.

10. Section 3.1:

• Table 1: The diameter of the inhibition zone of the negative control appears as 8 mm. Please check it carefully.

Response: Thanks to the reviewers for the reminder, The diameter of the Oxford cup used in the experiment was 8mm, and the measurement result of the negative control was 8mm, indicating that there was no inhibitory effect. Due to our incorrect expression, we have caused some trouble to the reviewers. For this problem, we have used "-" to indicate that there is no inhibitory effect. In addition, we specially marked the diameter information of the Oxford cup used in the experiment in the material method section

Please see lines 137.

Reviewer #2 (Comments for the Author):

I am favorably impressed with the novelty and character of this work. The initial review was taken constructively and my own recommendations have all been followed responsibly and professionally. I am confident about the professionalism and integrity displayed in the processing of this manuscript.

Response: *We sincerely appreciated the reviewer's review and positive comments.*

Others

1. We revised the title of the article as requested by the reviewer, and in this title we noticed the professional expression intestinal microbiota, so we replaced intestinal flora with intestinal microbiota in the whole article.

2. *lines 268-280:* In the previous comment, we noticed that the reviewers had requested to supplement the detailed steps of *PICRUSt* functional prediction, so we supplemented the detailed operation of *PICRUSt* functional prediction.

In addition, we checked the article data references and Checked for accuracy, Finally, thank the reviewers for their professional comments and patient guidance, I have made progress.

September 1, 2022

Dr. Chunyan Tan
Kunming University of Science and Technology
Yunnan , Kunming
China

Re: Spectrum01128-22R2 (Lactococcus lactis effect on the intestinal microbiota of Streptococcus agalactiae infected zebrafish (Danio rerio))

Dear Dr. Chunyan Tan:

Your manuscript has been accepted, and I am forwarding it to the ASM Journals Department for publication. You will be notified when your proofs are ready to be viewed.

Sincerely,

Konstantinos Kormas
Editor, Microbiology Spectrum
